
# MIPAS Observations of Ozone in the Middle Atmosphere

Manuel López-Puertas[1], Maya García-Comas[1], Bernd Funke[1], Angela Gardini[1], Gabriele P. Stiller[2], Thomas von Clarmann[2], Norbert Glatthor[2], Alexandra Laeng[2], Martin Kaufmann[3], Viktoria F. Sofieva[4], Lucien Froidevaux[5], Kaley A. Walker[6], and Masato Shiotani[7]

[1]Instituto de Astrofísica de Andalucía, CSIC, Granada, Spain
[2]Karlsruhe Institute of Technology, Institute of Meteorology and Climate Research, Karlsruhe, Germany
[3]Institute for Energy and Climate Research, Research Centre Jülich, Jülich, Germany
[4]Finnish Meteorological Institute, Earth Observation, Helsinki Finland
[5]Jet Propulsion Laboratory, California Institute of Technology, Pasadena, CA, USA
[6]6 Department of Physics, University of Toronto, Toronto, Ontario, Canada
[7]Research Institute for Sustainable Humanosphere, Kyoto University, Uji, Kyoto, Japan

*Correspondence to:* M. López-Puertas (puertas@iaa.es)

**Abstract.** In this paper we describe the stratospheric and mesospheric ozone (version V5r_O3_m22) distributions retrieved from MIPAS observations in the three middle atmosphere modes (MA, NLC and UA) taken with an unapodized spectral resolution of $0.0625 \, cm^{-1}$ from 2005 until April 2012. $O_3$ is retrieved from microwindows in the $14.8 \, \mu m$ and $10 \, \mu m$ spectral regions and requires non-LTE modelling of the $O_3$ $v_1$ and $v_3$ vibrational levels. Ozone is reliably retrieved from 20 km in the

MA mode (40 km for UA and NLC) up to $\sim$105 km during dark conditions and up to $\sim$95 km during illuminated conditions. Daytime MIPAS $O_3$ has an average vertical resolution of 3-4 km below 70 km, 6-8 km at 70–80 km, 8-10 km at 80–90 km and 5–7 km at the secondary maximum (90–100 km). For nighttime conditions the vertical resolution is similar below 70 km, and better in the upper mesosphere and lower thermosphere: 4-6 km at 70–100 km, 4–5 km at the secondary maximum, and 6–8 km at 100–105 km. The noise error for daytime conditions is typically smaller than 2% below 50 km, 2–10% between 50

and 70 km, 10-20% at 70-90 km and $\sim$30% above 95 km. For nighttime, the noise errors are very similar below around 70 km but significantly smaller above, being 10-20% at 75-95 km, 20-30% at 95-100 km and larger than 30% above 100 km. The additional major $O_3$ errors are the spectroscopic data uncertainties below 50 km (10-12%), and the non-LTE and temperature errors above 70 km. The validation performed suggests that the spectroscopic errors below 50 km, mainly caused by the $O_3$ air-broadened half-widths of the $v_2$ band, are overestimated. The non-LTE error (including the uncertainty of atomic oxygen

at nighttime) is relevant only above $\sim$85 km with values of 15–20%. The temperature error varies from $\sim$3% up to 80 km to 15-20% near 100 km. Between 50 and 70 km, the pointing and spectroscopic errors are the dominant uncertainties. The validation performed in comparisons with SABER, GOMOS, MLS, SMILES and ACE-FTS shows that MIPAS $O_3$ has an accuracy better than 5% at and below 50 km, with a positive bias of a few percent. In the 50-75 km region, MIPAS $O_3$ has a positive bias of $\approx$10%, which is possibly caused in part by $O_3$ spectroscopic errors in the $10 \, \mu m$ region. Between 75 and

90 km, MIPAS nighttime $O_3$ is in agreement with other instruments by 10%, but for daytime the agreement is slightly larger, $\sim$10-20%. Above 90 km, MIPAS daytime $O_3$ is in agreement with other instruments by 10%. At nighttime, however, it shows a positive bias increasing from 10% at 90 km to 20% at 95-100 km, the latter of which is attributed to the large atomic oxygen





abundance used. We also present MIPAS $O_3$ distributions as function of altitude, latitude and time, showing the major $O_3$ features in the middle and upper mesosphere. In addition to the rapid diurnal variation due to photochemistry, the data also show apparent signatures of the diurnal migrating tide, both during day and nighttime, as well as the effects of the semi-annual oscillation above $\sim$70 km in the tropics and mid-latitudes. The tropical daytime $O_3$ at 90 km shows a solar signature in phase
with the solar cycle.

## 1   Introduction

Ozone is a key constituent of the atmosphere playing a major role in its energy budget and chemistry, particularly in the stratosphere and upper mesosphere (Brasseur and Solomon, 2005). Typical ozone profiles show two clearly distinguished maxima, one located between 10 and 35 km, the so-called ozone layer, where most ozone resides, and a secondary maximum
around the mesopause ($\sim$90-95 km). Ozone has been extensively measured in the stratosphere and also in the mesosphere using different techniques (see, e.g. Kaufmann et al. (2003) for measurements taken prior to 2003 and Smith et al. (2013) for the most recent observations, mainly from satellite instruments). Measurements that cover the atmosphere from the lower stratosphere up to the lower thermosphere, both at day- and night-time conditions and with global latitudinal coverage are, however, not very frequent.

MIPAS (Michelson Interferometer for Passive Atmospheric Sounding) is a high-resolution limb sounder on board the Envisat satellite, launched on March 1, 2002 and taking measurements until 8 April 2012, when the Envisat satellite failed. It had a wide spectral coverage and a high spectral resolution, operating at 0.025 cm$^{-1}$ during 2002-2004 and 0.0625 cm$^{-1}$ from 2005 until the 8th of April 2012 (Fischer et al., 2008). MIPAS operated with a global latitude coverage (pole-to-pole) and performed measurements during day and night. The instrument spent most of the time observing in the 6-68 km altitude range
(the nominal mode, NOM) but it also regularly looked at higher altitudes in its middle atmosphere (MA), noctilucent (NLC), and upper atmosphere (UA) modes (De Laurentis, 2005; Oelhaf, 2008). The retrieval of ozone from the NOM mode has been carried out, among others, by the Institute of Meteorology and Climate Research and Instituto de Astrofísica de Andalucía (IMK/IAA) (Glatthor et al., 2006). In that inversion local thermodynamic equilibrium (LTE) is assumed, which is a good approximation up to about 50 km and a reasonable approach up to about 60 km (von Clarmann et al., 2009). MIPAS took a few
spectra of the middle atmosphere (only half a day of measurements on 11 June 2003) with its full spectral resolution mode. Ozone concentrations were retrieved from those spectra from the low stratosphere up to the lower thermosphere including the non-LTE effects (Gil-López et al., 2005). In this paper we focus on the inversion of ozone from the bulk of MIPAS observations of the middle atmosphere taken from 2005 until April 2012 in three different modes (MA, NLC and UA) with the optimized spectral resolution of 0.0625 cm$^{-1}$. Part of this dataset (2008-2009), version V4O_O3_m02, was previously
retrieved and used in some studies (e.g., Smith et al., 2013). Here we describe the inversion of the entire MA/UA/NLC period, 2005-2012, version V5r_O3_m22, derived from V5 L1b spectra. The method is essentially described in Gil-López et al. (2005). Here we review several aspects of the retrieval baseline including the changes in the non-LTE modeling of $O_3$ and the new microwindows (MWs) used. In addition, an assessment of the quality of the middle atmosphere $O_3$ data, a comparison with the



previous version V4O_O3_m02 (sometimes referred to as V4O_502), and a validation with recent middle atmosphere ozone measurements taken by SABER, GOMOS, MLS, SMILES and ACE-FTS, are also presented. Finally we describe the major features of the $O_3$ database focusing on the mesosphere.

## 2  $O_3$ non-LTE modelling

As mentioned below most of the microwindows used in the retrieval of $O_3$ in the mesosphere are located in the $9.6\,\mu m$ region, where the $v_1$ and $v_3$ fundamental, combinational and hot bands arise. We discuss here the non-LTE modelling of those vibrational levels. The $v_2$ fundamental band is used only at the lower altitudes, below $50\,km$, where it is in LTE (Funke et al., 2012). We should clarify that the non-LTE populations of $O_3$ used in the retrieval have been calculated with the Generic RAdiative traNsfer AnD non-LTE population algorithm (GRANADA) (see Sec. 3). In this section we describe a simplified formulation of non-LTE for $O_3$ with the aim of showing the quantities required for performing the retrieval.

The major processes affecting the populations of the vibrationally excited $O_3(v_1,v_2,v_3)$ levels are listed in Table 1. The collisional processes (1-4) have been adapted from Table 7 in Funke et al. (2012), to which we have added the simplified radiative processes of a) the absorption of radiation from the layers below (process 5) and b) the spontaneous emission (process 6).

**Table 1.** Collisional and radiative processes affecting the $O_3$ vibrational levels.

| No. | Rate | Process |
|-----|------|---------|
| 1 | $k_1$ | $O_2 + O + M \rightarrow O_3(v_1,v_2,v_3)$ |
| 2 | $k_{vt,M}$ | $O_3(v_1,v_2,v_3) + M \rightleftharpoons O_3(v'_1,v'_2,v'_3) + M$ |
| 3 | $k_{chem}$ | $O_3(v_1,v_2,v_3) + O \rightarrow O_2 + O_2$ |
| 4 | $k_{vt,O}$ | $O_3(v_1,v_2,v_3) + O \rightarrow O_3(v'_1,v'_2,v'_3) + O$ |
| 5 | $J_{Earth}$ | $O_3 + h\nu \rightarrow O_3(v_1,v_2,v_3)$ |
| 6 | $A$ | $O_3(v_1,v_2,v_3) \rightarrow O_3(v'_1,v'_2,v'_3) + h\nu$ |

Note that the energy of the $O_3(v_1,v_2,v_3)$ level is larger than that of $O_3(v'_1,v'_2,v'_3)$.

From the statistical equilibrium assumption and the processes listed in Table 1 the population of the vibrationally excited $O_3(v_3)$ levels (we focus here on $v_3$, from whose emission we are retrieving $O_3$) can be obtained, approximately, by

$$\frac{[O_3(v_3)]}{[O_3]} = \frac{J_{Earth} + p_t + p_{nt}}{A + k_{vt,M}[M] + k_{vt,O}[O] + k_{chem}[O]}, \tag{1}$$

where the brackets denote concentrations, $J_{Earth}$ is the excitation rate of $O_3(v_3)$ due to the absorption of $\nu_3$ photons coming from the lower atmospheric regions (upwelling flux), $A$ is the Einstein coefficient of the $\nu_3 = 1$ band, [O] the atomic oxygen concentration, [M] the air molecules concentration (sum of $N_2$ and $O_2$) and the other collisional rates are defined in Table 1. $p_t$ is the specific thermal production of $O_3(v_3)$ given by

$$p_t = k_{vt,M}[M]\exp[-E(v_3)/kT], \tag{2}$$





where $T$ is the kinetic temperature, $k$ the Boltzmann constant and $E(v_3)$ the energy of the $O_3(v_3)$ level. The specific production

due to non-thermal processes (chemiluminescence, process 1 in Table 1), $p_{nt}$, is given by

$$p_{nt} = k_1 \left[O_2\right]\left[M\right]\left(\left[O\right]/\left[O_3\right]\right)\phi(v_3), \tag{3}$$

where $\phi(v_3)$ is the fraction of the $O_3$ molecules produced by process 1 which finally end excited in the $O_3(v_3)$ level. In practice,

$\phi(v_3)$ depends not only on the nascent distribution of reaction 1 but also, and mainly for the lower $v_3$ levels, on the radiative

and collisional relaxations of the more energetic $(v_1,v_2,v_3)$ levels, that is on $A, k_{vt,M}, k_{vt,O}$ and $k_{chem}$ for those levels and also

on $[M]$ and $[O]$. From all those parameters, the only unknown in our case is the atomic oxygen concentration, $[O]$.

For the retrieval of $O_3$, one would like the $[O_3(v_3)]/[O_3]$ ratio of Eq. 1, which is proportional to the $O_3(v_3)$ vibrational

temperature and hence to the measured $O_3$ emission, to be the least dependent as possible on kinetic rates and other atmospheric

parameters. For the MIPAS $O_3$ retrieval, all the parameters in Eqs. 1, 2 and 3 are known, or measured simultaneously by MIPAS,

except $[O]$.

**Table 2.** Major photochemical reactions affecting $O_3$ in the mesosphere and lower thermosphere.

| No. | Rate | Process |
|-----|------|---------|
| 1 | $k_1$ | $O_2 + O + M \rightarrow O_3(v_1,v_2,v_3)$ |
| 2 | $k_2$ | $H + O_3 \rightarrow OH^*(v) + O_2$ |
| 3 | $k_3$ | $O + O_3 \rightarrow O_2 + O_2$ |
| 4 | $J_{Sun}$ | $O_3 + h\nu \rightarrow O_2 + O$ |

Because of the rapid timescales for ozone production and loss in the non-LTE region, the assumption of photochemical

equilibrium above around 60 km is valid. Thus, from the major photochemical reactions affecting $O_3$ in the mesosphere and

lower thermosphere (see Table 2), the non-thermal production, $p_{nt}$, and $[O]$ are given for daytime conditions, respectively, by

$$p_{nt,day} = J_{Sun}\,\phi(v_3)\quad\text{and} \tag{4}$$

$$[O]_{day} \approx \frac{J_{Sun}\,[O_3]}{k_1\,[O_2]\,[M]}, \tag{5}$$

where the chemical losses ($k_2$ and $k_3$) have been neglected. In this way, we have no major limitation of unknown atmospheric

parameters in the $O_3$ non-LTE retrieval during daytime if we update the non-LTE model with the $O_3$ abundance in each

iteration.

During nighttime, assuming also photochemical equilibrium and the processes in Table 2, the non-thermal production and

$[O]$ are given, respectively, by

$$p_{nt,night} = \frac{k_2[H]}{1 - \frac{k_3\,[O_3]}{k_1\,[O_2]\,[M]}}\quad\text{and} \tag{6}$$




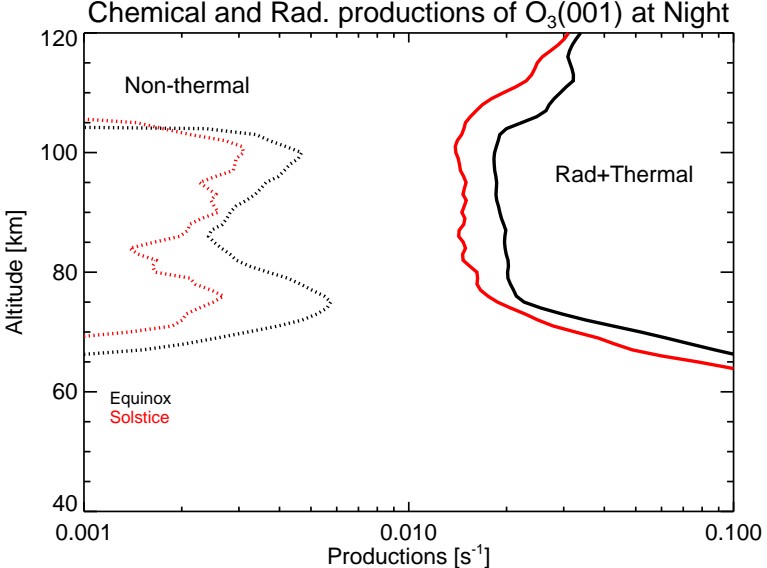

**Figure 1.** Specific production terms of the population of $O_3(001)$ at nighttime for two days typical of equinox (black) and solstice (red). Dotted lines are the specific non-thermal productions of $O_3(v_3=1)$ (Eq. 6), and solid lines are the sum of the radiative absorption and thermal collisions, $J_{Earth}$ and $p_t$ in Eq. 1.

$$[O]_{night} \approx \frac{k_2 [H] [O_3]}{k_1 [O_2] [M] - k_3 [O_3]}. \tag{7}$$

In this case, even iterating, we still have the unknown of the atomic hydrogen concentration, [H], which has to be taken from models or from parameterization. As we see, the non-thermal production at nighttime is directly proportional to [H]. Then, in

5  order to estimate the errors introduced by the uncertainty in [H], it is worthwhile to estimate how important is the contribution of the non-thermal term, $p_{nt,night}$ in Eq. 6, in comparison with the radiative and thermal contributions, $J_{Earth}$ and $p_t$ in Eq. 1. Figure 1 show the calculations of these terms computed with the GRANADA model for a few nighttime profiles for two days typical of equinox and solstice. The figure shows that the contribution of the non-thermal term to the excitation of $O_3(v_3=1)$ is, in the worst case, a factor of 5 smaller than the radiative and thermal excitations. We then expect that the uncertainties

10  in the inputed [H] from the NRLMSIS-00 model (Picone et al., 2002, see below) would not introduce large uncertainties in the retrieved $O_3$. This effect, however, is larger for higher $O_3$ vibrational levels because for these states the radiative term is negligible and the non-thermal part is larger. Thus the uncertainty in [H] might introduce a larger error for the case of wide band radiometers measuring a significant contribution of the hot bands emission.



**Table 3.** Microwindows and altitude ranges used in the retrieval of MIPAS ozone V5r_O3_m22.

| No. | Wavenumber ($cm^{-1}$) | | Altitudes (km) | | | | | | | | | | | | | | | | | | | | |
|---|---|---|---|---|---|---|---|---|---|---|---|---|---|---|---|---|---|---|---|---|---|---|---|
| | Minimum | Maximum | 18 | 21 | 24 | 27 | 30 | 33 | 36 | 39 | 42 | 45 | 48 | 51 | 54 | 57 | 60 | 63 | 66 | 69 | 72 | 75 | 78-102 |
| 1 | 687.6875 | 688.6875 | | | | | | ■ | | ■ | ■ | ■ | ■ | ■ | ■ | ■ | ■ | ■ | ■ | ■ | ■ | ■ | |
| 2 | 689.3125 | 691.8750 | | | ■ | ■ | ■ | | | ■ | ■ | ■ | ■ | ■ | ■ | ■ | ■ | ■ | ■ | ■ | ■ | ■ | |
| 3 | 692.2500 | 695.1875 | | | | | | | ■ | ■ | ■ | ■ | ■ | ■ | ■ | ■ | ■ | ■ | ■ | ■ | ■ | ■ | |
| 4 | 707.1250 | 710.0625 | | | | ■ | ■ | ■ | ■ | ■ | ■ | ■ | ■ | ■ | ■ | ■ | ■ | ■ | ■ | ■ | ■ | ■ | |
| 5 | 712.3125 | 713.4375 | | | | | | | | ■ | ■ | ■ | ■ | | ■ | | | | | | | | |
| 6 | 713.5000 | 716.4375 | | | ■ | ■ | ■ | ■ | ■ | ■ | ■ | ■ | ■ | ■ | | ■ | ■ | ■ | ■ | ■ | ■ | ■ | |
| 7 | 716.5000 | 719.4375 | | | ■ | ■ | | ■ | ■ | ■ | ■ | ■ | ■ | ■ | ■ | ■ | ■ | ■ | ■ | ■ | ■ | ■ | |
| 8 | 720.7500 | 723.6875 | | | | ■ | ■ | ■ | ■ | ■ | ■ | ■ | ■ | ■ | ■ | ■ | ■ | ■ | ■ | ■ | ■ | ■ | |
| 9 | 728.5000 | 729.3750 | | | | | | ■ | ■ | ■ | ■ | ■ | ■ | ■ | ■ | ■ | ■ | ■ | ■ | ■ | ■ | ■ | |
| 10 | 730.0625 | 730.5000 | | | | | | ■ | ■ | ■ | ■ | ■ | ■ | ■ | ■ | ■ | ■ | ■ | ■ | ■ | ■ | ■ | |
| 11 | 731.9375 | 732.8750 | | | ■ | ■ | ■ | | | ■ | | | | ■ | ■ | ■ | ■ | ■ | ■ | | ■ | ■ | |
| 12 | 734.0000 | 734.7500 | | | | | | ■ | ■ | | ■ | ■ | ■ | ■ | ■ | | | | | ■ | ■ | ■ | |
| 13 | 736.4375 | 739.3750 | ■ | ■ | | ■ | | ■ | ■ | ■ | ■ | ■ | ■ | ■ | ■ | ■ | ■ | ■ | ■ | ■ | ■ | ■ | |
| 14 | 739.4375 | 741.9375 | | | | | ■ | ■ | ■ | ■ | ■ | ■ | ■ | ■ | ■ | ■ | ■ | ■ | ■ | ■ | ■ | ■ | |
| 15 | 745.2500 | 745.6875 | ■ | | | | | ■ | ■ | ■ | ■ | ■ | ■ | ■ | ■ | ■ | ■ | ■ | ■ | ■ | ■ | ■ | |
| 16 | 746.6875 | 747.1250 | | ■ | ■ | ■ | ■ | ■ | ■ | ■ | ■ | ■ | ■ | ■ | ■ | ■ | ■ | ■ | ■ | ■ | ■ | ■ | |
| 17 | 747.6250 | 748.3750 | | | | ■ | ■ | ■ | ■ | | ■ | ■ | ■ | ■ | ■ | | | | ■ | ■ | ■ | ■ | |
| 18 | 749.5625 | 752.5000 | | ■ | ■ | ■ | ■ | ■ | ■ | ■ | ■ | ■ | ■ | ■ | ■ | ■ | ■ | ■ | ■ | ■ | ■ | ■ | |
| 19 | 752.9375 | 755.8750 | ■ | ■ | ■ | | | ■ | ■ | ■ | ■ | ■ | ■ | ■ | ■ | ■ | ■ | ■ | ■ | ■ | ■ | ■ | |
| 20 | 758.3750 | 759.4375 | ■ | ■ | ■ | ■ | ■ | ■ | ■ | ■ | ■ | ■ | ■ | ■ | ■ | ■ | ■ | ■ | ■ | ■ | ■ | ■ | |
| 21 | 759.5000 | 761.8750 | | | ■ | | | ■ | ■ | ■ | ■ | ■ | ■ | ■ | ■ | ■ | ■ | ■ | ■ | ■ | ■ | ■ | |
| 22 | 765.0000 | 765.6250 | | | | | | | ■ | ■ | ■ | ■ | ■ | ■ | ■ | ■ | ■ | ■ | ■ | ■ | ■ | ■ | |
| 23 | 767.5000 | 768.0000 | ■ | ■ | ■ | ■ | ■ | ■ | ■ | ■ | ■ | ■ | ■ | ■ | ■ | ■ | ■ | ■ | ■ | ■ | ■ | ■ | |
| 24 | 771.8750 | 772.1250 | ■ | ■ | ■ | ■ | ■ | ■ | ■ | ■ | ■ | ■ | | | ■ | ■ | ■ | ■ | ■ | ■ | ■ | ■ | |
| 25 | 774.2500 | 774.5625 | | ■ | ■ | ■ | | | ■ | ■ | ■ | ■ | | | ■ | | | | ■ | ■ | | | |
| 26 | 776.5000 | 776.7500 | | | | | | | | | ■ | ■ | | ■ | | ■ | ■ | ■ | ■ | ■ | | | |
| 27 | 780.2500 | 781.9375 | ■ | | | | | ■ | ■ | ■ | ■ | ■ | ■ | ■ | ■ | ■ | ■ | ■ | ■ | ■ | ■ | ■ | |
| 28 | 788.9375 | 789.6875 | ■ | | | | | ■ | ■ | ■ | ■ | ■ | ■ | ■ | ■ | ■ | ■ | ■ | ■ | ■ | ■ | ■ | |
| 29 | 790.7500 | 791.0000 | ■ | | | | | | | | | | ■ | | | | | | | | | | |
| 30 | 791.1875 | 791.5625 | ■ | ■ | | | | | ■ | ■ | ■ | ■ | ■ | ■ | ■ | ■ | ■ | ■ | ■ | ■ | ■ | ■ | |
| 31 | 1034.1250 | 1034.3750 | | | | | | | | | | | | | | | | | ■ | ■ | ■ | ■ | ■ |
| 32 | 1034.4375 | 1035.0000 | | | | | | | | | | | | ■ | ■ | ■ | ■ | ■ | ■ | ■ | ■ | ■ | ■ |
| 33 | 1038.1875 | 1039.0000 | | | | | | | | | | | | ■ | ■ | ■ | ■ | ■ | ■ | ■ | ■ | ■ | ■ |
| 34 | 1040.0000 | 1040.8125 | | | | | | | | | | | | ■ | ■ | ■ | ■ | ■ | ■ | ■ | ■ | ■ | ■ |
| 35 | 1048.8125 | 1049.5000 | | | | | | | | | | | | | | | | ■ | ■ | ■ | ■ | ■ | ■ |
| 36 | 1050.6250 | 1051.8125 | | | | | | | | | | | | | | | | ■ | ■ | ■ | ■ | ■ | ■ |
| 37 | 1053.3125 | 1053.8125 | | | | | | | | | | | | | | | | ■ | ■ | ■ | ■ | ■ | ■ |
| 38 | 1054.6875 | 1055.5000 | | | | | | | | | | | | | | | | ■ | ■ | ■ | ■ | ■ | ■ |

## 3 Ozone non-LTE retrieval

As described above, we used MIPAS spectra taken in the MA, NLC, and UA modes. In the MA mode, the spectra are taken at limb tangent heights from about 20 km up to 102 km with a vertical spacing of 3 km. The UA mode ranges from about 42 km up to 172 km, which has a 3 km vertical sampling up to 102 km and 5 km above. The NLC mode is a variant of the MA mode specifically tailored for measuring the NLCs during the summers (De Laurentis, 2005; Oelhaf, 2008). In this mode the spectra are taken at tangent heights from 39 km up to 78 km at 3-km steps; from 78 km up to 87 km at 1.5 km steps, and from 87 km up to 102 km again in 3-km steps. MIPAS horizontal field of view (FOV) is approximately 30 km.

The method used for the inversion of ozone under non-LTE conditions is essentially described in Gil-López et al. (2005). In that work the retrieval was adapted for the very few (only 26 orbits) MIPAS data of the upper atmosphere taken on June





2003 at the full spectral resolution of 0.025 cm$^{-1}$. Here we briefly summarize the approach, discuss the new MWs used in the inversion of the MA, NLC and UA observational modes (taken at 0.0625 cm$^{-1}$), and review the changes in the non-LTE modeling of $O_3$ vibrational levels.

The MIPAS V5r_O3_m22 ozone retrieval is based on a constrained multilinear least squares fitting of non-LTE limb radi-
ances. That is performed using the IMK/IAA Scientific Processor (von Clarmann et al., 2003, 2009) extended with the non-LTE GRANADA algorithm (Funke et al., 2012), which is able to cope with non-LTE emissions. Funke et al. (2001) describe the peculiarities of the retrievals under consideration of non-LTE. The basic retrieval equations, the methods for characterization of results through error estimates and vertical and horizontal averaging kernels, the iteration and convergence criteria and the regularization method are described in von Clarmann et al. (2003, 2009). Version V5 (5.02/5.06) of the ESA-calibrated L1b
spectra was used (see Raspollini et al., 2010, and references therein).

The inversion is performed after a retrieval of the residual spectral shift and the non-LTE retrieval of temperature (García-Comas et al., 2012; García-Comas et al., 2014). The IMK/IAA processor simultaneously retrieves, besides the $O_3$ abundance, microwindow- and altitude-dependent continuum radiation and zero level calibration corrections (the latter, assumed constant with altitude).

The retrievals are performed from the surface to 120 km over a fixed altitude grid of 1 km up to 50 km, at 72-75 km, and at 77-88 km; of 2 km at 50-72 km, 75-77 km, and 88-102 km; and of 5 km from 105 up to 120 km. The grids have been selected as a trade-off between higher accuracy and computational efficiency. The forward calculations are performed using the same grid. The over-sampled retrieval grid, finer than the MIPAS vertical sampling of 3 km, makes necessary the use of a regularization in order to obtain stable solutions. We used here a Tikhonov-type first order smoothing constraint (Tikhonov, 1963). The
numerical integration of the signal over the 3 km FOV is done using five pencil beams. The selected width of the integration window (apodized instrument line shape function) avoids channel border effects. Forward model calculations along the line of sight (LOS) are done considering horizontal gradient corrections. Thus, they account for changes in the populations (either in LTE or in non-LTE) of the emitting $O_3$ levels along the LOS due to kinetic temperature variations.

Ozone is reliably retrieved from 20 km in the MA mode (40 km for the UA and NLC modes) up to ∼105 km during dark
conditions and up to ∼95 km during illuminated conditions. The logarithm of the volume mixing ratio (vmr) is retrieved from MWs covering ro-vibrational emissions of the $O_3$ main isotope. They have been selected from a broad spectral region, covered by channel A (685–970 cm$^{-1}$), and channel AB (1020–1170 cm$^{-1}$), and they vary with tangent altitudes in order to optimize computation time and minimize systematic errors (von Clarmann and Echle, 1998). The MWs used are listed in Table 3. They are an extension of the set of MWs used for the ozone retrieved from the MIPAS NOM mode of observation, covering altitudes
below 70 km (von Clarmann et al., 2009). Below 50 km, we use the same MWs as in the NOM retrieval, located all in channel A. Above 50 km, the MWs used are mainly located in channel AB and are strongly affected by non-LTE.

The MIPAS V5r_O3_m22 ozone retrieval setup uses the following inputs. The $O_3$ a priori is taken from a MIPAS dedicated climatology (similar to that described in Funke et al., 2012). Pressure, LOS, temperature, and temperature horizontal gradients are taken from MIPAS T-LOS retrieval V5r_T_m21 (García-Comas et al., 2014). They have been retrieved from the $CO_2$ emis-
sion near 15 $\mu$m, recorded in channel A, and accounting for the non-LTE effects. The detailed description of the method and





the characterization of the inverted pressure-temperatures profiles are described in García-Comas et al. (2012). The upgrades in the retrieval of the temperature used here (V5r_T_m21) and a validation of the results are reported by García-Comas et al. (2014). This version of MIPAS temperatures correct the main systematic errors of the previous version and have, in general, a remarkable agreement with the measurements taken by ACE-FTS, MLS, OSIRIS, SABER, SOFIE and the Rayleigh lidars at

Mauna Loa and Table Mountain. In the region of interest here, however, there are still significant differences, with the MIPAS mesopause differing by 5-10 K from the other instruments, being warmer than SABER, MLS and OSIRIS and colder than ACE-FTS and SOFIE.

The $O_3$ spectroscopic data were taken from the HITRAN 2008 database (Rothman et al., 2009). A test performed using the HITRAN 2016 $O_3$ database (Gordon et al., 2017) has shown that the radiances are just marginally larger (only 0.5% in channel

A and 0.25% in channel AB).

The $O_3$ non-LTE vibrational populations were computed online in each iteration of the inversion by using the GRANADA model (Funke et al., 2012). As described above, the $O_3$ non-LTE model requires several inputs. The set of $O_3$ vibrational levels, the non-LTE collisional scheme and the rate constants are based on the GRANADA model except as noted below.

The exponent $a$ in the rate $k_1 = 6.0 \times 10^{-34} \, (T/300)^a$ of reaction $O + O_2 + M \rightarrow O_3(v_1, v_2, v_3) + M$ has been updated from

2.3 to 2.4 following Burkholder et al. (2015). This has a small effect on the $O_3$ $v_1$ and $v_3$ vibrational temperatures and also on the retrieved $O_3$.

The collisional deactivation of $O_3(v_1, v_2, v_3)$ (process 2 in Table 1) with $\Delta v_1$ or $\Delta v_3 = -1$ and $\Delta v_2 = 1$ (process 2c, $k_d$, in Table 7 of Funke et al. (2012)) was erroneously implemented in the previous version with a rate of $3.1 \times 10^{-15} \, (T/300)^{1/2}$. Now we include the expression in Table 7 of Funke et al. (2012) but limited to a minimum value of $4 \times 10^{-16}$ cm$^3$s$^{-1}$ at

temperatures below 200 K (Menard et al., 1992). Although this change in the rate coefficient is rather large, its impact on the $O_3(v_3)$ vibrational temperature is very small, as it is dominated by the relaxation to $v_2$. This leads to an $O_3$ vmr about 5% smaller between 70 and 85 km.

The chemical quenching of vibrationally excited $O_3$ by O, $O_3(v_1, v_2, v_3) + O \rightarrow O_2 + O_2$, has been neglected in this version. This implies larger $O_3(v_3)$ vibrational temperatures and hence smaller retrieved $O_3$ abundances. Note, however, that the total

quenching of $O_3(v_1, v_3)$ by O, including also collisional relaxation by O, process 4 in Table 1, is still within the measurement errors of the deactivation of $O_3(v_1, v_3)$ (West et al., 1976, 1978, see discussion below).

As shown in Sec. 2, the inversion of $O_3$ under non-LTE conditions requires knowledge of the atomic oxygen concentration, O. In our retrievals, we constrain the O abundance by employing photochemical equilibrium with the $O_3$ abundance retrieved in the previous iteration. This photochemical constraint for [O]/[$O_3$] is an optional feature in GRANADA which has been

switched on in this retrieval below 97 km.

At nighttime, the non-thermal production and O depends not only on $O_3$ but also on hydrogen, H (see Eqs. 6 and 7). For this reason, we use the H abundance from the NRLMSIS-00 model (Picone et al., 2002) in the photochemical equilibrium computation. Above 97 km we use the atomic oxygen from the Whole Atmosphere Community Climate Model with specified dynamics (SD-WACCM) simulations spanning over the time period of the measurements (Garcia et al., 2014). To be more

precise both profiles were merged in the 92-102 km region using a hyperbolic tangent function centred at 97 km. SD-WACCM



(in the following just WACCM) is constrained with output from NASA's Modern-Era Retrospective Analysis (MERRA) (Rienecker et al., 2011) below approximately 1 hPa. The main reason for including WACCM about that altitude is motivated by the lack or incorrect latitudinal and seasonal variation of atomic oxygen in the NRLMSIS-00 model.

During the inversion process we also include in the calculation of the spectra the contribution (as a potential overlap with $O_3$ lines) of the $CO_2$ lines. As for $O_3$, we also include the emissions by $CO_2$ bands in non-LTE. A detailed description of the $CO_2$ non-LTE model used and all the required input parameters can be found in Jurado-Navarro et al. (2016).

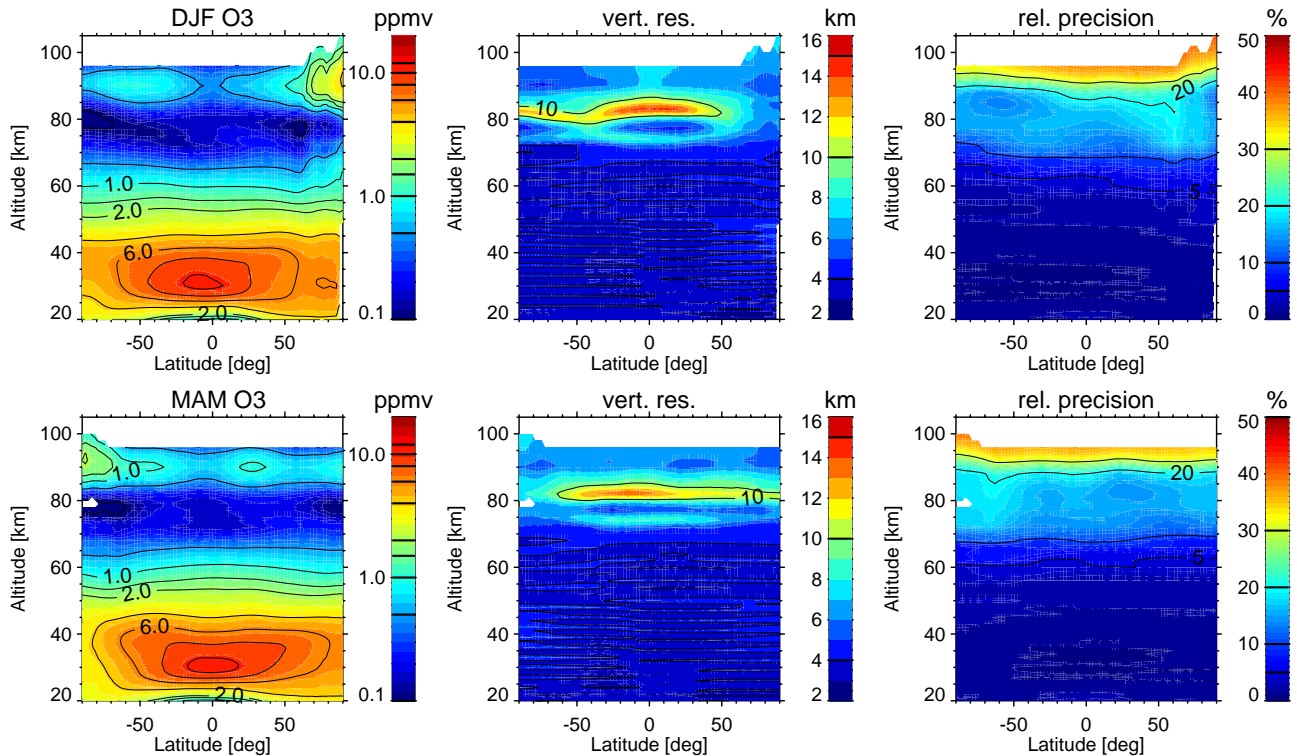

**Figure 2.** Latitude-altitude cross sections of MIPAS MA daytime ozone (left), its vertical resolution (center) and single profile noise error (1-$\sigma$, right) for solstice (December-January-February: DJF) (top row) and equinox (March-April-May; MAM) (bottom row). All measurements from 2005 to 2012 are included. White areas denote regions where the retrieved $O_3$ is not significant. Contour lines are marked in the color bar scale.

## 4 Characterization of the retrieved $O_3$ and error analysis

Figures 2 and 3 show seasonal zonal means of $O_3$ retrieved from MIPAS and the mean of the noise error (1-$\sigma$) and vertical resolution for the middle atmosphere (MA) mode for daytime (10 am) and nighttime (10 pm) conditions, respectively. The results for the UA and NLC modes are very similar except that the $O_3$ is retrieved above 40 km instead of 20 km. The ozone fields are included in these figures only for a reference to the noise and vertical resolution; its major features and the vertical





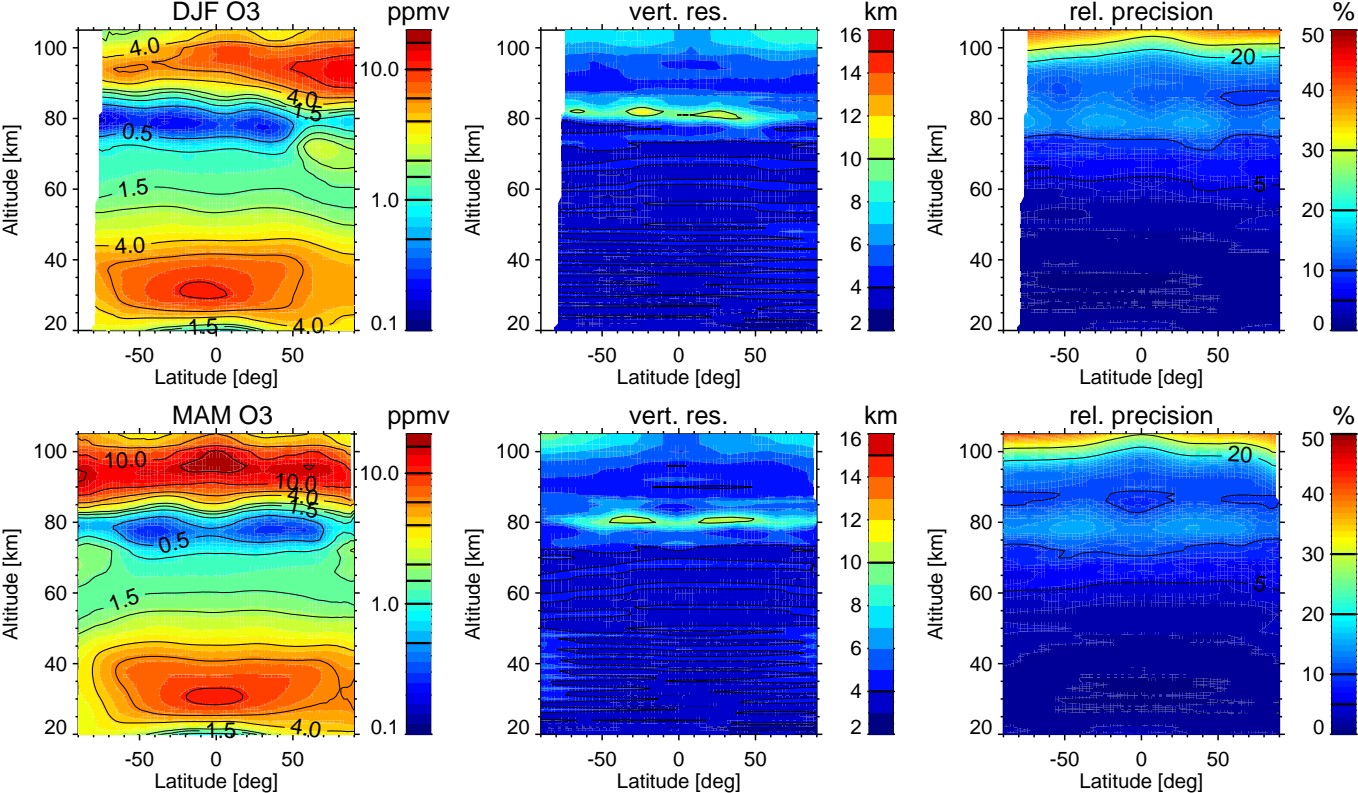

**Figure 3.** As Fig. 2 but for nighttime conditions.

and latitude distributions are discussed in Sec. 7 below. Two seasons are shown; the noise error and vertical resolution for the other two seasons are very similar.

The vertical resolution of the MIPAS retrieved ozone is given by the full width at half maximum of the averaging kernels rows. For daytime, the $O_3$ average vertical resolution is 3-4 km below 70 km, 6-8 km at 70–80 km, 8-10 km at 80–90 km

5 (although it can be coarser in tropical regions), and 5–7 km at the secondary maximum (90–100 km). For nighttime conditions the vertical resolution is similar below 70 km, and it is better in the upper mesosphere and lower thermosphere. It is 4-6 km at 70–100 km (except a narrow region near 80 km where it takes values of 8-10 km), 4–5 km at the secondary maximum; and 6–8 km at 100–105 km.

Two criteria are recommended to be used to screen the retrieved $O_3$ version V5r_O3_m22 data in order to guarantee the

10 quality of the profiles: a) the retrieved $O_3$ values of individual profiles where the diagonal value (or the mean diagonal value when averaging) of the averaging kernel is less (in absolute value) than 0.03 are considered non-trustful, and b) those values corresponding to altitudes not sounded by MIPAS (below the lowermost tangent altitude) and flagged by the visibility flag should not be used.





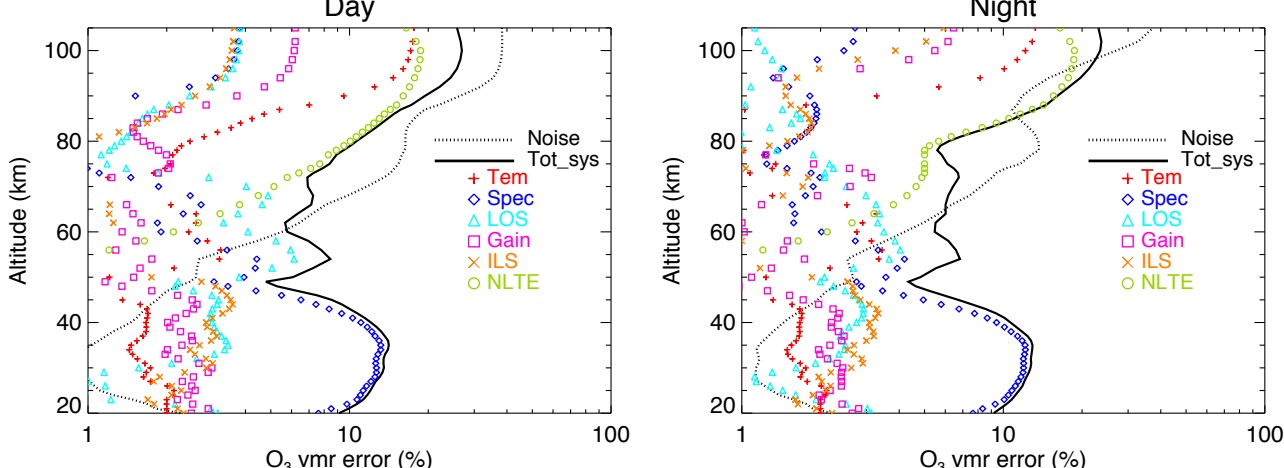

**Figure 4.** Errors of the retrieved $O_3$ vmr for daytime (left) and nighttime (right) in relative values. The different sources are described in the legend of Table 4.

The error budget described here considers the propagation of the measurement noise and of the uncertainties of model parameters onto the retrieved ozone abundances. Noise-induced retrieval errors (as well as the vertical resolution, see above) are estimated routinely for each individual profile by the retrieval algorithm. The ozone noise error is calculated assuming a wavelength-dependent noise-equivalent-spectral-radiance which has approximated average values of 17 and 10 nW/(cm$^2$cm$^{-1}$sr) for channels A and AB, respectively. Typical values (1-$\sigma$) for daytime (see the right column of Fig. 2) are smaller than 2% below 50 km, 2-10% between 50 and 70 km, 10-20% at 70-90 km and about 30% above 95 km. For nighttime (see right column of Fig. 3), the noise errors are very similar below around 70 km but significantly smaller above, being 10-20% at 75-95 km, 20-30% at 95-100 km and larger than 30% above 100 km.

**Table 4.** Summary of main errors (1-$\sigma$) of ozone vmr in %. Errors refer to nighttime conditions with daytime values, when different, in parentheses. 'NLTE' includes errors due to uncertainties in the collisional rates of the non-LTE model and in the atomic oxygen (see text). 'Total (Syst.)' is the root sum square of all systematic errors (noise is not included). LOS stands for Line Of Sight, 'Tem' for temperature, ILS for Instrument Line of Shape, and 'Spec.' for spectroscopy.

| Height (km) | Noise | LOS | Gain | Tem | ILS | Spec. | NLTE | Total (Syst.) |
|---|---|---|---|---|---|---|---|---|
| 100 | 27 (38) | 1.2 (3.8) | 5.5 (6.2) | 12 (17) | 4.0 | 2.5 (3.7) | 19 | 24 (27) |
| 95 | 17 (36) | 1.4 (3.5) | 2.1 (5.6) | 9 (15) | 1.8 (3.3) | 1.4 (3.3) | 18 | 21 (25) |
| 90 | 12 (26) | 1.0 (2.5) | 0.32 (3.7) | 3.3 (9.5) | 1.5 (2.6) | 1.8 | 16 | 16 (19) |
| 80 | 14 (15-20) | 0.8 (1.3) | 0.34 (1.6) | 0.8 (2.5) | 1.5 (0.9) | 1.6 (0.6) | 6 (10) | 6 (11) |
| 70 | 8.2 (12) | 2.5 (4) | 2.6 (0.9) | 0.4 (0.7) | 1.7 (0.7) | 1.9 | 5 | 6.4 (6.8) |
| 60 | 4.2 (5.5) | 3.5 (4.1) | 1.0 (1.4) | 2.8 (2.4) | 0.7 (0.9) | 2.0 | 2.0 | 5.5 (5.7) |
| 50 | 2.7 | 2.1 (3.8) | 1.1 (1.5) | 1.2 | 1.9 | 3.6 (4) | 0 | 4.8 (6) |
| 40 | 1.5 | 2.7 | 2.2 | 1.7 | 3.1 | 9.4 (11) | 0 | 11 (12) |
| 30 | 1.1 | 1.6 | 2.4 | 1.8 | 2.6 | 12 | 0 | 13 |
| 20 | 2.2 | 3.1 | 2.6 | 2.0 | 2.2 | 7.6 | 0 | 9 |



Errors related to the mapping of uncertain model parameters on the retrieved ozone vmr are estimated for representative profiles for daytime and nighttime conditions. We name these "systematic" errors to distinguish them from the noise error, although they are not purely systematic but in some cases they also have a random component. Fig. 4 shows those errors $(1\text{-}\sigma)$, including also mean profiles of the noise errors shown in Figs. 2 and 3, and Table 4 lists them. The uncertainties assumed for

the estimation of the systematic errors are 1% for gain calibration and 3% for the instrument line shape (ILS). For the elevation pointing (LOS) we have assumed an error of 150 m below 60 km, where we have information on the relative pointing from the temperature-LOS retrieval (von Clarmann et al., 2009; García-Comas et al., 2012). Above that height, where we obtain the LOS information from the engineering tangent altitudes of MIPAS (adjusted with the LOS retrieved below), we assumed an error of 300 m. For temperature, the errors have been considered as 0.5 K below 50 km, 1 K at 50-70 km, 2 K at 70-80 km, 5 K

at 80-100 km and 10 K above (García-Comas et al., 2014).

The spectroscopic errors have been estimated by J.-M. Flaud (personal communication, 2008). The errors assumed for the line intensities of the fundamental $v_1$, $v_2$ and $v_3$ bands (those mainly used here) are 2% for the strongest lines, 5% for the moderate lines, and 10% for the weaker lines. For the lines of the hot bands having as lower state (010), (020), (100) or (001), the error is in the range of 4-25%, increasing as the line intensity decreases. The errors considered for the more excited hot

bands are larger, up to 30%, but their contributions to the selected MWs is small. The error assumed for the $O_3$ air-broadened half-widths is 12%. The larger $O_3$ spectroscopic errors below 50 km (see Fig. 4) are mainly due to the error in the half-widths. The contribution of the error in the line intensities is of only 2–3%.

The modelling of the non-LTE populations of the $O_3$ vibrational levels emitting near 9.6 μm is an important source of the MIPAS ozone systematic error above the mid-mesosphere. Non-LTE errors are dominated by the uncertainties in the collisional

rates used in the non-LTE model. The three major sources are: 1) The error in the three-body reaction rate of $O_3$ formation; 2) the thermal relaxation of the $O_3(v_1,v_3)$ levels with $N_2$ and $O_2$; and 3) the collisional relaxation and/or chemical reaction of $O_3(v_1,v_3)$ levels with atomic oxygen. According to the current literature, we have considered uncertainties of 10% for the first, 20% for the second, and 50% for the third. For the later, West et al. (1976) derived experimentally the deactivation rate of $O_3(v_1,v_3)$ by O, and provided rates for both the thermal relaxation, $O_3(v_1,v_3) + O \rightarrow O_3 + O$, and the chemical

reaction, $O_3(v_1,v_3) + O \rightarrow 2\,O_2$, assuming each one alone was responsible for the total deactivation. These authors could not estimate the relative contribution of each process and the error associated with these two rates is about 50%. West et al. (1978) further reported that the deactivation is most likely to occur through the thermal relaxation. Hence, we have considered that the deactivation takes place through the thermal relaxation (process 2 in Table 1) and assumed the measured error of West et al. (1976) of 50%.

We have not considered additional errors due to the uncertainty in the atomic oxygen (daytime) or atomic hydrogen (nighttime) below 95 km (the error due to the $O_3$ error itself is already taken into account). During daytime, this is reasonable because atomic oxygen is derived from the photochemical equilibrium with the retrieved $O_3$. At night, however, when the retrieval depends on the atomic hydrogen concentration, if the error in H is significantly smaller than the 50% assumed for the deactivation of $O_3(v_1,v_3)$, its contribution will not be significant. If comparable, however, we might be underestimating the non-LTE errors.

Above 97 km, we have assumed an uncertainty in the WACCM atomic oxygen of 50%, leading to an error of about 10% in




the retrieved $O_3$ in that region (see Fig. 5). This error has been added, quadratically, to the other components of the non-LTE uncertainties discussed above.

The uncertainty of the first of those three processes on $O_3$ is only important above $\sim$85 km, reaching a maximum of 5% in the retrieved $O_3$. The second one is significant only from $\sim$60 km up to $\sim$80 km, introducing an error of 2–5% in ozone for

5    nighttime, and of 2–10% for daytime, including polar summer, where it might be slightly larger near 80 km. The third one, including the error in the atomic oxygen, is only significant above $\sim$85 km but it is the largest, with an estimated error in $O_3$ of 15–20% at 85-100 km. Overall, the non-LTE errors are typically negligible below 60 km, 2-10% at 60-80 km, and 15–20% above 85 km.

The overall systematic component of the $O_3$ abundance error is dominated by the spectroscopic data uncertainty below

10    50 km and by the non-LTE and temperature errors above about 70 km. Between 50 and 70 km, the pointing (LOS) and the spectroscopic errors are the dominant uncertainties. Our validation studies suggests, however, that the spectroscopic errors below 50 km are overestimated (see Sec. 6).

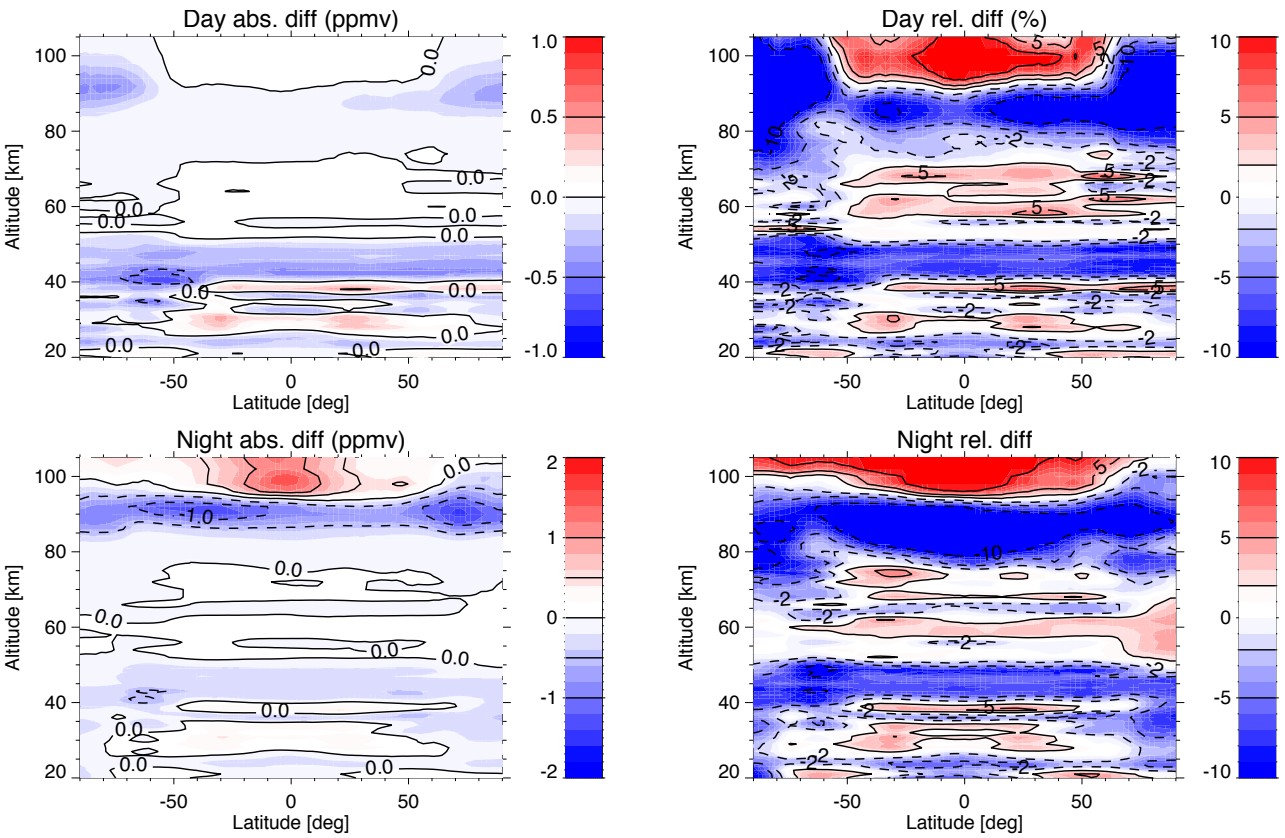

**Figure 5.** Comparison of $O_3$ abundance retrieved in the current V5r_O3_m22 with the previous V4O_O3_m02 version for daytime conditions (10 am) (top) and nighttime (bottom). The plots show the mean of the differences, in % of the older version, (V5r_O3_m22-V4O_O3_m02)/V4O_O3_m02, for all data taken in 2009.



## 5 Differences between current V5r_O3_m22 and previous V4O_O3_m02 versions

Since the previous V4O_O3_m02 version of MIPAS $O_3$ has been used in some previous studies, (e.g., Smith et al., 2013, 2014) it is interesting to compare those results with the vmrs of this new V5r_O3_m22 version. The V5r_O3_m22 ozone retrieval setup has been improved with respect to the earlier in the following aspects.

First, the MIPAS L1b spectra have been updated from version 4.61/62 to version V5 (5.02/5.06). The spectroscopic data were upgraded from the HITRAN database version of 2004 to that of 2008. The retrieved kinetic temperature from MIPAS has also been changed to the temperature version of v5r_TLOS_m21 (see major differences discussed above). We have also improved the width of the integration window of the apodized instrument line shape function. The uppermost altitude of the continuum retrieval has been expanded from 30 to 50 km. The regularization scheme has been updated in order to make it compatible

with that used in the nominal $O_3$ retrieval (von Clarmann et al., 2013) in the common altitude range (up to 70 km). The MWs have also been revised so below 50 km we only use MWs located in channel A (see Table 3), like in NOM retrievals. We use a different distribution of the $CO_2$ abundance, now taken from WACCM (see above). The merging altitude of the daytime atomic oxygen derived from the ozone retrieval (assuming photochemical equilibrium) to the model (supplied for the region above) has been changed from 95 km to 97 km. At nighttime we need, additionally, the H concentration, which has been taken from

the NRLMSIS-00 model (Picone et al., 2002). In the previous version we took the nighttime O from the NRLMSIS-00 model in the whole altitude range. Above 97 km we use now the atomic oxygen from WACCM instead of that of NRLMSIS-00. An important update is the re-calculation of the $O_3$ vibrational temperatures during the iterations in the inversion process, following the update of the atomic oxygen. The $O_3$ non-LTE model has also been improved, as described above in Sec. 2.

The average impact on the ozone retrieval after those changes (see Fig.5) is an increase of 2-3% (0.2-0.5 ppmv) below

around 40 km (except in the polar winter where there is a decrease). The clear increase around 40 km is due to the inclusion of the retrieval of the continuum for this altitude.

There is a clear decrease by ∼5-10% (∼0.4-0.5 ppmv) between 40 and 50 km principally induced by the use of micro-windows in channel A only (MWs in channel AB for these altitudes were removed). In the lower and middle mesosphere, 50-80 km, there is an increase of about 2-5% (0.1-0.2 ppmv). In the upper mesosphere, there is a general decrease of ∼0.5-1

ppmv at nighttime conditions, principally caused by neglecting the removal of the excited $O_3(v_3)$ by chemical reaction with atomic oxygen. This produces a larger population of $O_3(v_3)$ and hence less $O_3$. At daytime the effect is much smaller and has a smaller impact on the retrieved ozone (in absolute values). At altitudes above around 95 km, the $O_3$ in the new version is larger by about 5-10%. This is caused by the use of the atomic oxygen from the WACCM model, which is larger than in NRLMSIS-00 and even overcomes the smaller relaxation of $O_3(v_3)$ due to the chemical relaxation being ignored.

## 6   Validation

We have compared MIPAS V5r_O3_m22 ozone retrievals with co-located measurements from SABER, GOMOS, MLS, SMILES and ACE-FTS. Comparisons for GOMOS are only for night conditions and in number density. For ACE-FTS, because it is an occultation instrument and $O_3$ has very large diurnal variations around the terminator in the middle and upper



mesosphere, we compare ACE sunset and sunrise with MIPAS observations with solar zenith angles (SZA) in the range of 88 to 92°. In order to select a pair of profiles to compare, we have selected measurements with Universal Time differences smaller than 2 h and distances smaller than 1000 km. Considering an additional criterion of 1 h local time difference did not change the results significantly.

## 6.1 Instruments

### 6.1.1 SABER

The Sounding of the Atmosphere using Broadband Emission Radiometry (SABER) is a broadband radiometer flying onboard the NASA's Thermosphere-Ionosphere-Mesosphere Energetics and Dynamics (TIMED) satellite, launched on December 2001 and starting operations in January 2002 (Russell III et al., 1999). SABER measurements cover 83°S to 52°N and from 52°S to 83°N, alternatively every two months. A 24-h local time coverage is completed approximately in 60 days. SABER observes the daytime and nighttime ozone limb emission at $9.6\,\mu$m, from which the ozone concentration is retrieved under non-LTE conditions from 10 to 100 km. We use here version 2.0 ozone, publicly available at http://saber.gats-inc.com. The non-LTE model used in SABER ozone retrievals is described in Mlynczak et al. (2013) and references therein. Similar to MIPAS, the retrieval of $O_3$ from SABER requires knowledge of pressure, temperature and atomic oxygen. The first two are taken from SABER retrievals of simultaneous measurements at $15\,\mu$m (Remsberg, 2008; García-Comas et al., 2008). The atomic oxygen for SABER ozone retrievals is taken from the US Naval Research Laboratory- Mass Spectrometer and Incoherent Scatter Radar model (Picone et al., 2002). SABER ozone retrieval additionally needs to include the contribution of one $CO_2$ laser band emission in the ozone channel. For that it uses the $CO_2$ vibrational temperatures computed during the SABER temperature retrieval. The vertical resolution of SABER ozone is approximately 2 km. Given MIPAS $O_3$ coarser vertical resolution, particularly in the mesosphere, we used the MIPAS averaging kernels and a priori $O_3$ to smooth SABER $O_3$ profiles. Rong et al. (2009) reports SABER ozone precision of $\approx 1-2\%$ in the stratosphere and $\approx 3-5\%$ in the lower mesosphere. The systematic errors range from 22% in the lower stratosphere to $\approx 10\%$ in the lower mesosphere.

Previous comparisons between SABER v1.07 and MIPAS V4O_O3_m02 ozone vmrs in the mesosphere were performed by Smith et al. (2013). These showed the largest differences at the secondary maximum, which were attributed to the coarser MIPAS vertical resolution. They also mentioned that differences arose from the different SABER and MIPAS pressure/temperature profiles which affected conversion from density to vmr. The differences between SABER version 1.07 used in Smith et al. (2013) and version 2.0 (used here) are small (Smith et al., 2014).

### 6.1.2 GOMOS

The Global Ozone Monitoring by Occultation of Stars (GOMOS) was a stellar occultation spectrometer onboard the ESA's Envisat space platform (Bertaux et al., 2010). It operated from August 2002 to April 2012. Ozone density profiles are derived from GOMOS UV-visible measurements at $250-692$ nm from 10 to 110 km. GOMOS provides nighttime observations that are performed at around $22-23$h LST at low latitudes. The latitudinal coverage eventually reaches the poles and slightly varies throughout the year due to varying distribution of stars used in the observations. The dataset used here was retrieved



using ESA Instrument Processor Facility (IPF) version 6.01, described in Kyrölä et al. (2010) and Sofieva et al. (2010), and is available under registration at the ESA Earth online portal (https://earth.esa.int). Unreliable profiles in the dataset have been screened out following recommendations of the GOMOS/6.01 Level 2 Product Quality Readme file. The user-friendly version of the GOMOS dataset (HARMOZ, Sofieva et al., 2013), which is screened for invalid data, is in an open access at the

Ozone_cci web-page, http://www.esa-ozone-cci.org/?q=node/161. The vertical resolution of GOMOS ozone varies from 2 km in the lower stratosphere to 3 km in the upper stratosphere and above. Being better than the MIPAS vertical resolution, we have used MIPAS averaging kernels to smooth GOMOS profiles. Since GOMOS provides $O_3$ number density but not $O_3$ vmr we compare GOMOS and MIPAS $O_3$ number densities. Random errors due to measurement noise and scintillations are $0.5-4\%$ in the stratosphere and $2-10\%$ in the mesosphere. Systematic errors are smaller than 2% and they are mainly due to $O_3$ cross

sections (Tamminen et al., 2010).

### 6.1.3   MLS

The Microwave Limb Sounder (MLS) was launched on July 2004 on the NASA's Earth Observing System Aura satellite (Waters et al., 2006). The equatorial crossings occur at 1:43/13:43 LT and the latitudinal coverage is between 82°S and 82°N. Daytime and nighttime ozone profiles are derived from measurements of its thermal limb emis-

sion at 235.71 GHz from the troposphere to ≈90 km. However, its usable range is up to 0.02 hPa or ∼72 km (see https://mls.jpl.nasa.gov/products/o3_product.php). The ozone dataset used here is version 4.2, downloaded from GES DISC (Schwartz et al., 2015) and described in Livesey et al. (2017) and references therein. The vertical resolution is 3 km in the stratosphere, 6 km in the middle mesosphere and 9 km in the upper mesosphere. MLS ozone has generally indicated 5-10% agreement with other datasets in the stratosphere. The estimated systematic uncertainty is 5-10% in the stratosphere, 10-20% in

the lower mesosphere and 20-50% in the middle mesosphere Livesey et al. (2017). Due to the larger MLS $O_3$ vertical resolution in the mesosphere, we have applied MLS averaging kernels and a priori information to the MIPAS ozone.

### 6.1.4   SMILES

The Superconducting Submillimeter-Wave Limb-Emission Sounder (SMILES) was attached to the Exposed Facility of the JAXA's Japanese Experiment Module (JEM) of the International Space Station (ISS) and operated between October 2009 and

April 2010 (Kikuchi et al., 2010). It measured ozone profiles from 16 to 85 km during daytime and to 96 km during nighttime, derived from measurements between 625 and 651 GHz using the technique described in Mitsuda et al. (2011), Takahashi et al. (2010), and Takahashi et al. (2011). The latitudinal coverage is 38°S and 65°N. We use here data version 3.2, available at the Data Archives and Transmission System (DARTS) site (http://darts.isas.jaxa.jp/stp/smiles/). The vertical resolution is 3 km in the stratosphere, 4 km in the lower and mid-mesosphere and 6 km at 95 km. Systematic errors range between 5 and 10% above

30 km (Baron et al., 2011). MIPAS and SMILES $O_3$ vertical resolutions are similar and hence we did not apply averaging kernels to any of them.

Previous versions of SMILES $O_3$ (v2.2) agree with other measurements within 10% in the stratosphere and 30% in the mesosphere (Imai et al., 2013a, b).



### 6.1.5 ACE-FTS

The Fourier Transform Spectrometer (ACE-FTS) is an infrared solar occultation Michelson interferometer flying on the CSA's Atmospheric Chemistry Experiment (ACE), also called Science Satellite (SciSat), launched in August 2003 (Bernath, 2017). It measures atmospheric absorption from the cloud top to 150 km during sunrise and sunset. The ACE orbit's high inclination

results in coverage of the tropical, mid-latitude and high-latitude regions over approximately three months. Ozone profiles are derived from measurements at several microwindows between 829 and 2673 $cm^{-1}$ using the methodology described in Boone et al. (2013). We use here data version 3.5. The retrievals are limited to the altitude range of 5 to 95 km. The vertical resolution is $3-4$ km. The ACE-FTS version 3.5 $O_3$ product agrees with MLS measurements to within +4% from 20-45 km. Compared to MLS, it exhibits a positive bias of up to 18% between 45 and 60 km and an increasing negative bias at higher altitudes (Sheese

et al., 2017).

### 6.2   Results of comparisons

Figures 6 and 7 show the mean daytime and nighttime differences, respectively, between MIPAS and the different instruments (MIPAS−instrument) for the four seasons, grouped in four latitude bins. In addition, Figs. 8 and 9 show the global means, for all latitudes and seasons, for daytime and nighttime, respectively. The number of co-located pairs included in each mean

difference profile is indicated as a subindex in the instrument's label. The mean altitude of the MIPAS $O_3$ vmr primary and secondary maxima, coincident with the respective instrument, are also plotted as additional information.

    In general, the agreement with all instruments, except SABER, is better than 5% below 50 km in all seasons for both daytime and nighttime; MIPAS $O_3$ being larger (see Figs. 8 and 9). The differences around the ozone primary maximum (diamonds in the figures) are smaller than 5%. These differences are well within MIPAS systematic errors (see Table 4). However, MIPAS

(as the other instruments) measures less ozone than SABER below 50 km, with values of 10-20% from 30 to 50 km and 5-15% at the stratospheric $O_3$ maximum.

    At altitudes from 50 to 65-70 km, we also find a good general agreement with all instruments, except SABER at 60–70 km, and MLS at 65–70 km in some latitudes/seasons. The differences are smaller than 5-10% in all seasons both during daytime and nighttime, with MIPAS $O_3$ generally being larger (see Figs. 8 and 9). These differences are within or just at the edges of the

MIPAS systematic errors (grey shading in the figures). Opposite to that behaviour, MIPAS is smaller (5–10%) than ACE-FTS in the altitude range of 45–55 km (see Fig. 8). This is likely the effect of the known positive bias of ACE-FTS $O_3$ in this region (Sheese et al., 2017). MIPAS measures up to 10% (nighttime) and up to 20% (daytime) less ozone than SABER from 50 to 60 km. These differences increase above 60 km and are discussed below.

    Exceptions to that general behavior are the larger differences found with MLS at 65-70 km at daytime in spring (at 30°-50°)

and autumn and winter for most latitudes. Differences with SMILES are also exceptionally larger at 60–80 km during nighttime in winter at 70°-90° but, with only 8 coincidences, they are not statistically significant.

    A positive bias in SABER stratospheric $O_3$, version 1.07, was previously reported by Rong et al. (2009). Differences in temperature cannot explain this bias because they are 1–2 K larger than MIPAS below 30 km (which would result in less $O_3$)





**Figure 6.** Mean of the daytime $O_3$ vmr differences (MIPAS–instrument) in % of MIPAS between co-located pairs of measurements of MIPAS (MA mode) with ACE-FTS (green), SABER (red), MLS (purple), and SMILES (magenta) for spring (MAM for NH and SON for SH) (upper/left quadrant), autumn (SON for NH and MAM for SH) (upper/right quadrant), summer (JJA for NH and DJF for SH) (bottom/left quadrant), and winter (DJF for NH and JJA for SH) (bottom/right quadrant). The symbols indicate the mean altitude of the MIPAS $O_3$ vmr primary (diamonds) and secondary (circles) maxima coincident with the respective instrument. The numbers of coincidences are indicated in the subscripts. The grey shaded area shows the MIPAS systematic errors. The color-shaded areas (hardly noticeable in many cases) are the standard errors of the mean of the differences.



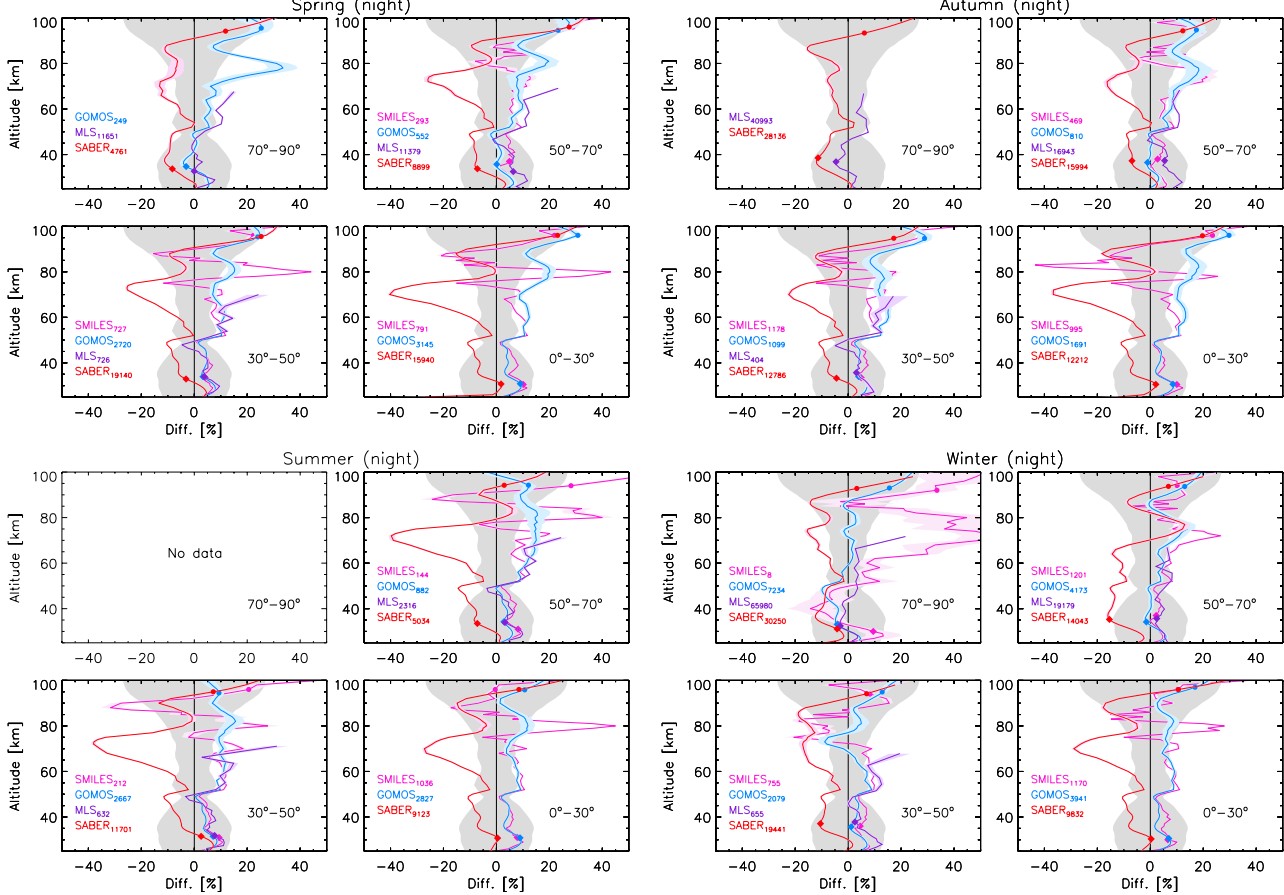

**Figure 7.** As Fig. 6 but for nighttime $O_3$. Instruments colors are the same except that ACE-FTS is replaced by GOMOS (light blue).

and are in excellent agreement (within 1 K) from 30 to 85 km (García-Comas et al., 2014). Some tests have shown the retrieval of MIPAS $O_3$ at these altitudes using MWs in the AB channel, e.g. the 10 $\mu$m spectral region where SABER measures, results in an 10% ozone increase (see Sec. 5 and Fig. 5). Thus, $O_3$ spectroscopic errors in the 10 $\mu$m region could be the reason for the larger SABER stratospheric ozone. Indeed, MIPAS and SABER measurements have a better agreement at 50 km (altitude

5 above which MIPAS uses AB channel MWs) than at lower altitudes, whereas MIPAS differences with instruments other than SABER increase to 10% above that altitude.

Between 50 and 85 km MIPAS and SABER ozone profiles show similar vertical gradients but shifted (not shown). In this region ozone decreases with altitude to very small values and hence the relative differences are rather large. At 60–85 km they are within 20–80% at daytime, and 10–40% at nighttime, MIPAS ozone being smaller. A daytime SABER ozone overestimation

10 at these altitudes was already reported by Smith et al. (2013). A likely explanation for the larger SABER $O_3$ values could be the faster de-activation of the $O_3$ $v_1$ and $v_3$ manifold by $N_2$ and $O_2$, $k_{vt,M}$ in Table 1. According to Mlynczak et al. (2013), the SABER retrieval uses the values reported by Martin-Torres (1999) which are about a factor of 2 faster than those measured



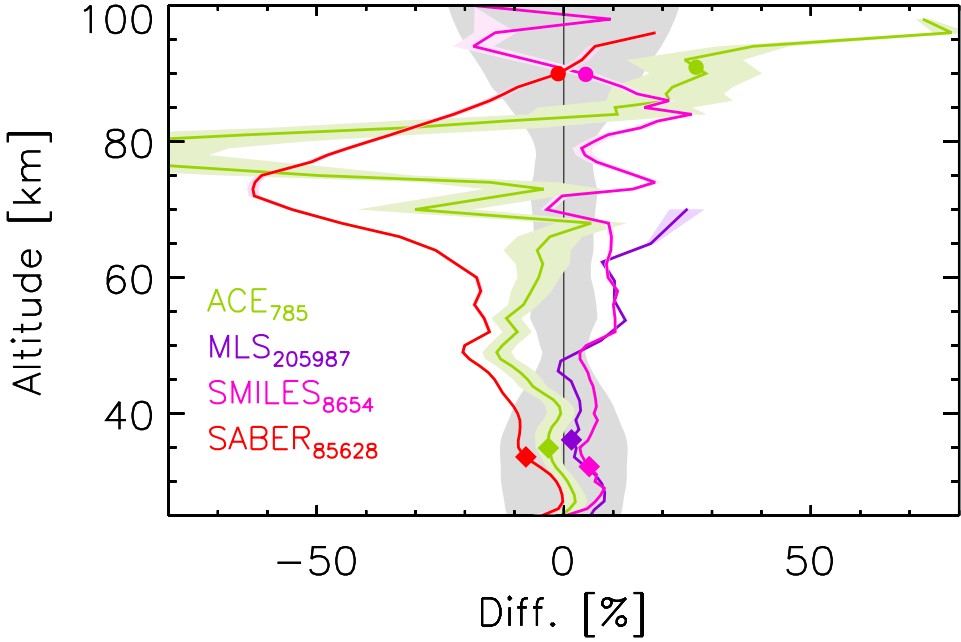

**Figure 8.** Global mean (for all latitudes and seasons) of the daytime $O_3$ vmr differences (MIPAS–instrument) in % of MIPAS between co-located pairs of measurements of MIPAS (MA mode) with ACE-FTS (green), SABER (red), MLS (purple), and SMILES (magenta). For more details see caption of Fig. 6.

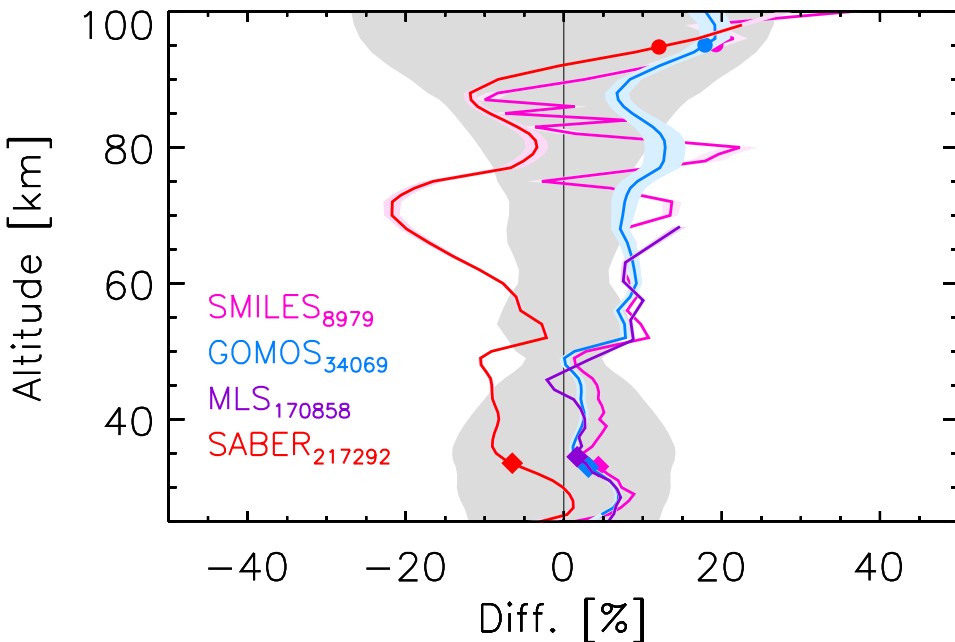

**Figure 9.** As Fig. 8 but for nighttime $O_3$. Instruments colors are the same except that ACE-FTS is replaced by GOMOS (light blue).



by Menard et al. (1992) which are used here. A faster collisional rate gives rise to smaller $O_3(v_3)$ vibrational temperatures and hence to larger retrieved $O_3$ vmr. The fact that the non-LTE deviation of $O_3(v_3)$ vibrational temperatures from the kinetic temperature is larger at daytime than at nighttime could explain the larger daytime bias.

MIPAS differences with other instruments are larger above 70 km than below. These differences are of 10–20% with SMILES and GOMOS (nighttime). When compared with ACE-FTS, MIPAS $O_3$ is significantly smaller ($<-50\%$) near 80 km (see Fig. 8). This large relative difference is in part caused by the very small values of $O_3$. The difference in absolute values is smaller than 0.1 ppmv. Also, the difference could be due to different solar illuminations along the LOS of both instruments.

In the polar winter around 70 km, where the tertiary maximum develops (see Sec. 7), MIPAS agreement with GOMOS is excellent (3%) (see Fig. 11, winter at 70-90°). MIPAS and GOMOS lie between the rest of the other instruments, being on average 10% smaller than SABER (night), 20% larger than MLS (during night), within ∼30% of ACE-FTS (terminator) and 50% larger (although with a very few coincidences) than SMILES (night). Smith et al. (2013) reported that SABER version 1.07 $O_3$ did not exhibit a tertiary maximum, which contrasts with version 2.0 where SABER shows the largest ozone tertiary maximum of all the instruments considered.

At altitudes above 85–90 km during nighttime the differences increase with altitude and are generally larger. In the case of MIPAS and SABER the differences are partially due to a vertical shift of 1–2 km (not shown).

At the altitude of the secondary maximum, the agreement between all instruments for daytime conditions, is very good, except ACE-FTS in autumn and winter. MIPAS daytime ozone differences with SABER and SMILES are within 5 and 10%, respectively, except for low latitudes in the solstices (10%) and summer high latitudes (20%) for SABER, and autumn high latitudes and winter low latitudes (20%) for SMILES. MIPAS ozone is generally larger than that of ACE (by a mean of ∼20–30%) at this altitude. This difference can be partially due to the different illumination conditions along the LOS of both instruments. Actually we performed a test, restringing MIPAS data to SZA$<85°$, and the difference was reduced to just 10% (0.2-0.3 ppmv).

During nighttime, however, MIPAS ozone is between 5% and 25% larger than the other instruments around the secondary maximum. The differences somehow vary with season, being largest in spring and autumn. MIPAS ozone is 10–20% larger than SABER and SMILES, and 10-25% larger than GOMOS.

At altitudes above around 93 km, this larger MIPAS nighttime ozone is most likely caused by the large atomic oxygen in WACCM. If using the MSIS atomic oxygen in this region those differences would be reduced by 10–15% (see Figs. 6 and 7).

Overall, focusing on Figs. 8 and 9, MIPAS $O_3$ has an accuracy better than 5% at and below 50 km, with a positive bias of only a few percent. In the 50-75 km region, MIPAS $O_3$ has a positive bias of approximately 10%, possibly caused by spectroscopic errors. Between 75 and 90 km, nighttime MIPAS $O_3$ is within 10% of SABER and SMILES $O_3$ but has a positive bias of about 10% with respect to GOMOS. In this region, during daytime, the relative differences are larger, with a positive difference of 10–20% with SMILES, and negative difference of 10–50% with ACE and SABER. Given that ACE measures at the terminator and SABER daytime $O_3$ has a known positive bias, we think that MIPAS $O_3$ in this region is accurate to within 10-20%. Above 90 km, MIPAS $O_3$ at daytime is in agreement with other instruments by 10%. At nighttime, however, it shows a positive bias increasing from 10% at 90 km to 20% at 95-100 km, which is attributed to the large atomic oxygen of WACCM.







**Figure 10.** Composite monthly zonal mean of MIPAS data taken in the MA mode for the 2007-2012 period for daytime (local time of 10 am). White areas denote regions where MIPAS has no sensitivity to measure the very low ozone values. Contours are 0.1, 0.5, 1, 1.5, 2, 4, 6, 8, 10 and 12 ppmv.





**Figure 11.** Composite monthly zonal mean of MIPAS data for the 2017-2012 period for nighttime. Contours are 0.5, 1, 1.5, 2, 4, 6, 8, 10, 15, and 20 ppmv.





# 7   Climatology

The MIPAS middle atmosphere $O_3$, with a global (pole-to-pole) latitude coverage, day and nighttime measurements taken at two fixed local times, and spanning the altitude range from 20 to 100 km, represents a very important dataset for studying the middle atmosphere. In this section we present some ozone distributions, represented versus altitude, latitude and time, showing
the major $O_3$ features in the middle and upper mesosphere.

Figures 10 and 11 show composite monthly zonal means of MIPAS $O_3$ data for the 2007-2012 period for day- and nighttime, respectively. Here we consider "daytime" the measurements with a solar zenith angle (SZA) smaller than the SZA of the terminator, $SZA_{ter}(z) = 180 - \arcsin(R_\oplus/(R_\oplus + z))\,180/\pi$ ($R_\oplus$ is the Earth's radius and $z$ is altitude) decreased in 3°, and taken at 10 am local time. The decrease in 3° was done to avoid scans partially in dark conditions. "Nighttime" is taken when
the observed altitude is in dark conditions, $SZA(z) > SZA_{ter}(z)$, and the measurements are taken at a local time of 10 pm. Those figures show the typical primary, secondary and tertiary maxima and their seasonal evolution.

The latitudinal/seasonal distribution of daytime $O_3$ in the secondary maximum, near 90-95 km, shows maxima near the polar winters (November-February in the NH, and May-August in the SH) (Fig. 10).

In general we observe a minimum in the daytime $O_3$ secondary maximum near the tropics (except perhaps for October)
which is attributed to tidal effects (see, e.g., Marsh et al., 2002; Dikty et al., 2010). The diurnal tertiary maximum, taking place usually around 60-75 km (Fig. 10), occurs during the winter seasons polewards of 60-70°.

The ozone secondary maximum at nighttime presents larger values near the polar winters, more precisely in the early polar winters (e.g. November in the NH and May in the SH). This polar winter maximum decreases as the season progresses in both hemispheres, and starts recovering near the end of the winter (February in the NH and August in the SH). This is in agreement
with the results reported by Smith et al. (2014). These investigators have shown that this $O_3$ enhancement is caused by the relatively weak meridional circulation at those times/regions which leads to low temperatures and low H concentrations, both favouring the production of $O_3$.

Thus, during solstice, the $O_3$ secondary maximum shows a clear latitudinal gradient, growing from summer to winter. During the equinox months, the nighttime secondary maximum exhibits large $O_3$ values across all latitudes, reaching the highest values
in April and October. In the equinox months of March, April, September and October, the signature of the diurnal migrating tide is apparent near the equator. Larger values are found near the equator than at adjacent latitudes near 75-80 km, then smaller around 85-87 km, and larger again near 95 km. The signature is more clearly seen in April and October when the largest values of the ozone secondary maximum are observed at the equator near 95 km, reaching up to about 20 ppmv. The nighttime tertiary maximum has larger values than during daytime (see Fig. 11) and it is usually shifted equatorwards. These values are in good
agreement with the measurements of the nighttime tertiary maximum measured by GOMOS (Sofieva et al., 2009). At latitudes poleward of the terminator it is usually displaced to lower altitudes. Although not clearly noticeable (because of the scale) in Fig. 11, the nighttime tertiary maximum is slightly larger in the NH than in the SH, as reported by Smith et al. (2017).







**Figure 12.** Seasonal evolution versus latitude of $O_3$ vmr at different altitudes for daytime (left row) and nighttime (right row). Note the different scales used in the different panels.



## 7.1 Annual variability

Figure 12 shows the annual variability as latitude× months cross sections of $O_3$ at different altitudes for daytime (left column) and nighttime (right column).

At 40 km MIPAS observes larger $O_3$ values in the tropics and mid-latitudes compared to the polar regions, and also lower values in the SH polar winter compared to the NH winter. Around the equator we observe slightly larger values during solstice conditions (January and July) than at equinox (April and October). Maximum values also occur at mid-latitudes (30–40°) during May-June and September in the SH and during August and September in the NH. The day-night differences are small, with ozone slightly larger during daytime, more noticeable in the NH.

At 60 km the day-night differences are very marked, with smaller daytime ozone values due to losses by photo-dissociation. This pattern, due to illumination conditions, is very clear in the second panel of the first column.

At 70 km, the structure of the $O_3$ tertiary maximum (second column, third panel) in the SH polar winter is very obvious, exhibiting a "ring" shape following the terminator. The tertiary maximum has larger values in the early winter, then decreases and increases again at the end of the winter. Note also the higher values of the tertiary maximum in the NH polar winter (as discussed above). These results are consistent with the recent analysis of the tertiary maximum carried out by Smith et al. (2017).

The $O_3$ latitude × month distribution near 80 km shows during equinox conditions significant increases at mid-latitudes during daytime and in the tropics at nighttime, which seem to be caused by tides. Near the polar regions the nighttime $O_3$ at 80 km shows similar features to the tertiary maximum described above with larger values early in the winter in both hemispheres, and being larger in the NH.

At 90 km, the nighttime $O_3$ in the polar winter shows a more marked seasonal evolution than at lower altitudes, with much larger values in the early winter as has been shown by Smith et al. (2014). The semi-annual oscillation (SAO) is also evident in the tropics and at mid-latitudes (see, e.g., Garcia et al., 1997). Similar features have also been observed by GOMOS (Kyrölä et al., 2010). Even though $O_3$ has a short lifetime, and hence it is not significantly advected, its distribution responds to changes in temperature, atomic oxygen, and other species associated with the SAO. As a result of both effects ozone shows a kind of "U-shape" distribution at this altitude.

## 7.2 Altitude-resolved time series

In order to study the inter-annual variability We show in this section altitude-resolved time series of $O_3$ at the tropical and polar latitudes (Fig. 13); and as latitude×time cross sections at given altitudes (Fig. 14). We see clearly the semi-annual oscillation above around 75 km, both in day and nighttime in the upper panels of Fig. 13. We also observe in these panels that the $O_3$ secondary maximum is located at higher altitudes in nighttime than during daytime. In addition, there is a hint that the daytime $O_3$ secondary maximum shows the lowest concentrations close to the solar cycle minimum in 2009/2010, which is also visible in the bottom/left panel of Fig. 14.





**Figure 13.** Altitude-resolved series, for latitudes near the equator (10°S-10°N) (top), the Southern polar region, 70°S-90°S, (middle) and the Northern polar region, 70°N-90°N (bottom), for daytime (left column) and nighttime (right column).



**Figure 14.** Cross section of latitude/time MIPAS $O_3$ at 50 km, 70 km and 90 km for daytime (left column) and nighttime (right column). Note the different color scales in some panels.



Focusing on the polar regions (middle and bottom panels of Fig. 13), we observe that the stratospheric maximum (25-40 km) is slightly larger in the NH polar region than in the SH polar region. As discussed above, the nighttime tertiary maximum is slightly larger in the NH than in the SH polar region. In the SH it shows a double peak in each winter (early and late in the winter) while in the NH it is not so pronounced. Moving to higher altitudes, the nighttime $O_3$ secondary maximum shows a

clear winter variability, with larger values early in the winter. The double peak structure appearing each winter is very evident in the SH hemisphere, as it is in the tertiary maximum, but not so clear in the NH hemisphere.

Figure 14 shows that the diurnal variation is clearly seen already at 50 km at the tropics and mid-latitudes, with slightly smaller values at daytime due to losses by photo-dissociation. This diurnal variation increases with altitude and at 70 km it is already quite large.

The presence of the tertiary maximum near 70 km at the polar regions leads to larger $O_3$ vmr values at these latitudes than at tropical and mid-latitudes, opposite to the latitudinal gradient shown at 50 km. In the middle/right panel of Fig. 14 we also observe that the nighttime tertiary maximum near 70 km is larger in the NH than in the SH hemisphere, as discussed above.

As described above, the bottom/left panel shows that the tropical daytime $O_3$ at 90 km shows lowest concentrations close to the solar cycle minimum in 2009/2010. This behavior is in concordance with model simulations (Marsh et al., 2007) and

is explained by decreased odd oxygen production via $O_2$ photolysis at low solar activity conditions. The opposite occurs, however, at 70 km (middle/left panel), with a tendency to decrease towards 2012. Observational evidence for a negative solar ozone response at these altitudes has been provided by the analysis of Solar Mesosphere Explorer (SME) data on solar rotation time scales (Keating et al., 1987). Our observations suggest a long-term decline rather than a solar cycle variation since no $O_3$ increase before the solar minimum in 2009 can be identified. However, a clear attribution is not possible due to the relatively

short observation period of MIPAS.

## 8    Summary and Conclusions

In this paper we describe the stratospheric and mesospheric ozone distributions (version V5r_O3_m22) retrieved from MIPAS observations in the three middle atmosphere modes (MA, NLC and UA) taken with an unapodized spectral resolution of 0.0625 cm$^{-1}$ from 2005 until April 2012. The non-LTE modelling of $O_3$ is described in detail with emphasis on the unknown

atmospheric and model parameters required to retrieve ozone from limb emission measurements near 10 $\mu$m. In particular we discuss the role of atomic oxygen, which can be obtained during daytime by assuming it is in photochemical equilibrium with the retrieved $O_3$, but it is a source of uncertainty at nighttime.

We succinctly described the retrieval method and update the $O_3$ non-LTE model parameters and the new microwindows.

Regarding the quality of the retrieved MIPAS $O_3$, for daytime it has an average vertical resolution of 3-4 km below 70 km,

6-8 km at 70–80 km, 8-10 km at 80–90 km and 5–7 km at the secondary maximum (90–100 km). For nighttime conditions the vertical resolution is similar below 70 km, and it is better in the upper mesosphere and lower thermosphere. It is 4-6 km at 70–100 km (except a narrow region near 80 km where it is coarser), 4–5 km at the secondary maximum, and 6–8 km at 100–





105 km. We recommend to use MIPAS $O_3$ only when the absolute value of the diagonal (or the mean diagonal when averaging) of the averaging kernel is larger than 0.03.

The ozone noise error for daytime is typically smaller than 2% below 50 km, 2–10% between 50 and 70 km, 10-20% at 70-90 km and ∼30% above 95 km. For nighttime, the noise errors are very similar below around 70 km but significantly smaller

above, being 10-20% at 75-95 km, 20-30% at 95-100 km and larger than 30% above 100 km.

The major $O_3$ parameter errors are the spectroscopic data uncertainties below 50 km (10-12%) and the non-LTE and temperature errors above about 70 km. The validation analysis points to differences versus other datasets that are well within the estimated systematic uncertainties.

The non-LTE error (including the uncertainty of atomic oxygen at nighttime) is significant only above ∼85 km with values

of 15–20%. The temperature error varies from ∼3% near 80 km to 15-20% near 100 km. Between 50 and 70 km, the pointing and the spectroscopic errors are the dominant uncertainties.

The ozone of this version shows, compared with the previous V4O_O3_m02 version (V4O_502 in some papers), an increase of 2-3% (0.2-0.5 ppmv) below around 40 km (except in the polar winter where it is smaller); a decrease of ∼5-10% (∼0.4-0.5 ppmv) between 40 and 50 km (due to the use of MWs of channel A only); and an increase of about 2-5% (0.1-0.2 ppmv) at

50-80 km. In the upper mesosphere, there is a general decrease of ∼0.5-1 ppmv at nighttime, principally caused by neglecting the removal of $O_3(v_3)$ by chemical reaction with atomic oxygen. Above around 95 km, the $O_3$ in the new version is larger in about 5-10% due to the use of the larger atomic oxygen from the WACCM model.

The validation performed in comparisons with SABER, GOMOS, MLS, SMILES and ACE-FTS, shows that MIPAS $O_3$ has an accuracy better than 5% at and below 50 km, with a positive bias of a few percent. In that region, MIPAS systematic

errors, mainly caused by the $O_3$ air-broadened half-widths of the $v_2$ band, seem to be overestimated. In the 50-75 km region MIPAS $O_3$ has a positive bias of approximately 10%, which is possibly caused in part by $O_3$ spectroscopic errors in the 10 μm region. Between 75 and 90 km, MIPAS nighttime $O_3$ is in agreement with other instruments by 10%. At daytime, in this region, because of the low $O_3$ values, the error can be ∼10-20%. Above 90 km, MIPAS $O_3$ at daytime is in agreement with other instruments by 10%. At nighttime, however, it shows a positive bias increasing from 10% at 90 km to 20% at 95-100 km,

which is attributed to the large abundances of atomic oxygen of the WACCM model.

The systematically larger $O_3$ measured by SABER below 50 km when compared with all instruments considered here, including MIPAS, which in this region uses the $v_2$ band near 14.8 μm but not the 10 μm bands used by SABER, suggests that there might be a problem in the spectroscopic data of the $O_3$ 10 μm bands (or another unknown problem).

The global latitude coverage together with measurements at two fixed local times makes MIPAS $O_3$ very useful for studying

the seasonal changes and partially its diurnal variation, globally in both hemispheres. The major features are summarised here. The latitudinal/seasonal distribution of daytime $O_3$ in the secondary maximum, near 90-95 km, shows maxima near the polar winters where the SZA is rather large and losses by photo-dissociation smaller. Near the tropics it exhibits a minimum which is attributed to tidal effects (see, e.g., Marsh et al., 2002; Dikty et al., 2010).

MIPAS $O_3$ data also shows the typical tertiary maximum, taking place around 60-75 km in the winter seasons at latitudes

polewards of 60-70°. Its extension and magnitude are larger in the middle of the winters, and exhibit a hemispheric asymmetry



with larger values in the NH. Furthermore, it has larger values during nighttime when it is usually shifted equatorwards. These features are in concordance with the results reported by Smith et al. (2017).

Note that $O_3$ is higher in these regions in the late fall to early winter (November-December in the NH and May-June in the SH), similar to the results reported by Smith et al. (2014). These investigators have shown that this $O_3$ enhancement is caused

by the relatively weak meridional circulation at those times/regions which leads to low temperatures and low H concentrations, both favouring the production of $O_3$.

The ozone nighttime secondary maximum presents the largest values in the early polar winters, decreases as the season progresses and recovers near the end of the winter. This is in concordance with the results reported by Smith et al. (2014). Thus, during solstice, the $O_3$ nighttime secondary maximum shows a latitudinal gradient growing from summer to winter.

During equinox, the nighttime secondary maximum exhibits an apparent signature of the diurnal migrating tide, reaching the highest vmrs (up to about 20 ppmv) at the equator near 95 km.

MIPAS $O_3$ upper mesospheric data also exhibit the effects of the semi-annual oscillation (SAO) in the tropics and mid-latitudes.

The tropical daytime $O_3$ at 90 km shows the lowest concentrations close to the solar cycle minimum in 2009/2010; in

concordance with model simulations (Marsh et al., 2007) and explained by decreased odd oxygen production via $O_2$ photolysis at low solar activity conditions. At 70 km, however, one observes an opposite effect, with a tendency for ozone decreases towards 2012.

Observational evidence for a negative solar ozone response near 70 km has been provided by the Solar Mesosphere Explorer (Keating et al., 1987). MIPAS data suggest, however, more of a long-term decline than a solar cycle variation, although a clear

attribution is not possible due to the relatively short observation period of MIPAS.

*Acknowledgements.* The IAA team was supported by the Spanish MICINN under project ESP2014-54362-P and EC FEDER funds. The IAA and IMK teams were partially supported by ESA $O_3$-CCI and MesosphEO projects. MGC was financially supported by MINECO through its 'Ramón y Cajal' subprogram. Funding for the Atmospheric Chemistry Experiment comes primarily from the Canadian Space Agency. Work at the Jet Propulsion Laboratory was performed under contract with the National Aeronautics and Space Administration.





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
