# Peer review of "MIPAS Observations of Ozone in the Middle Atmosphere"

_Atmospheric Measurement Techniques, 2017_

## Referee Comment (RC1) · Anonymous Referee #1 · 16 Jan 2018

This manuscript describes a version 5 MIPAS ozone dataset for 2005-2012, as based on its optimized spectral resolution measurements. It is a comprehensive and very well written document. I recommend one addition. Please show one or more (daily or monthly), zonal-average pressure/latitude cross sections of day minus night (or vice versa) ozone (in %) for the stratosphere through lower mesosphere (your LTE region). I have not seen this kind of diagnostic in any previous publication about the MIPAS ozone dataset. Such a plot can be an important internal check about the registration of the radiance profiles and of the retrieved ozone and temperature profiles from this particular infrared, limb-emission experiment. If such diagnostic plots also look good, they would signal to me that your LTE ozone and temperature profile dataset must be of good quality. Of course, there will be more uncertainty for ozone in the upper

mesosphere, where NLTE processes are important and where you must make use of unmeasured [O] and [H] distributions in your retrieval algorithms.

---

## Referee Comment (RC2) · Anonymous Referee #2 · 25 Feb 2018

This paper describes the characteristics and validity of the new data product of MIPAS O3 measurements, including the comparison with other satellite data and also the climatology of O3 distribution. The manuscript is well prepared and is recommended for the publication in AMT.

Below I put some minor comments from the view point of a better readability, particularly for those who is not so familiar with this kind of satellite remote sensing.

-p.2 Line 15: Please clarify the local times of MIPAS observation (or, please say something about the sun synchronized orbit of Envisat). Although such information is provided at the beginning of Sec.4, I think it is still useful for readers to know it in advance at this introduction section.

[Figure]

-p.3 Section 2: I would suggest to include relevant references about the general intro-
duction of the Non-LTE processes of O3.

-p.6 Table 3 shows the micro-windows that the authors used in the retrieval. I would
like to see an example of L1b spectra at several tangent heights. This gives us an idea
about how low the radiance noise is (which the authors describes in page 11 line 5).

-p.7 Line 10: What is the major improvement of the new version 5 of L1b spectra
compared to the previous one (particularly compared to v-4.61/62)?

-p.9 Line 4: "...the calculation of the spectra the contribution (as a...": this sentence
appears odd.

-p.10 Line 11: Threshold of the averaging kernel 0.03, is this an empirical value?

-p.12 Error analysis for the systematic errors: I would suggest to add a short descrip-
tion about how the authors evaluated the systematic errors (numerically computed by
comparing the retrieved profiles using the nominal inversion model and the modified
inversion model?).

-p.16 Line 30: Baron et al. (2011) seems a reference paper for the earlier version of
SMILES O3 data.

---

## Author Comment (AC1) · 9 Mar 2018

**Response to the comments of Reviewer #1**

We are glad to hear that the reviewer found the manuscript as a comprehensive document. The reviewer comments are given below in blue and our responses in black.

This manuscript describes a version 5 MIPAS ozone dataset for 2005-2012, as based on its optimized spectral resolution measurements. It is a comprehensive and very well written document. I recommend one addition. Please show one or more (daily or monthly), zonal-average pressure/latitude cross sections of day minus night (or vice versa) ozone (in %) for the stratosphere through lower mesosphere (your LTE region). I have not seen this kind of diagnostic in any previous publication about the MIPAS ozone dataset. Such a plot can be an important internal check about the registration of the radiance profiles and of the retrieved ozone and temperature profiles from this particular infrared, limb-emission experiment. If such diagnostic plots also look good, they would signal to me that your LTE ozone and temperature profile dataset must be of good quality. Of course, there will be more uncertainty for ozone in the upper mesosphere, where NLTE processes are important and where you must make use of unmeasured [O] and [H] distributions in your retrieval algorithms.

Even though the manuscript has already many figures, we have included an additional figure with zonal mean day-night differences of MIPAS O3 for the four seasons. The reviewer suggested to show the differences only in the stratosphere and lower mesosphere, the LTE region, however, since the space required in the manuscript is the same we have shown the differences for the whole altitude of the retrieval. This gives a better overall view of the differences and might then be useful for more readers. In effect, the differences found in MIPAS are in line with previous measurements and current model predictions and hence render confidence on the good quality of MIPAS O3 data. We have included a short paragraph just before Sec. 7.1 describing the figure and making that statement.

---

## Author Comment (AC2) · 9 Mar 2018

**Response to the comments of Reviewer #1**

We thank the reviewer for the careful reading and constructive comments. The reviewer comments are given below in blue and our responses in black.

This paper describes the characteristics and validity of the new data product of MIPAS O3 measurements, including the comparison with other satellite data and also the climatology of O3 distribution. The manuscript is well prepared and is recommended for the publication in AMT.
Below I put some minor comments from the view point of a better readability, particularly for those who is not so familiar with this kind of satellite remote sensing.

-p.2 Line 15: Please clarify the local times of MIPAS observation (or, please say something about the sun synchronized orbit of Envisat). Although such information is provided at the beginning of Sec.4, I think it is still useful for readers to know it in advance at this introduction section.

Done. Information on the Envisat orbit, altitude and local times has been inserted.

-p.3 Section 2: I would suggest to include relevant references about the general introduction of the Non-LTE processes of O3.

Done. A new reference, a book on non-LTE, has been introduced in the 2nd. par. of Sec. 2.

-p.6 Table 3 shows the micro-windows that the authors used in the retrieval. I would like to see an example of L1b spectra at several tangent heights. This gives us an idea about how low the radiance noise is (which the authors describes in page 11 line 5).

We were not sure about including a figure since the paper is already rather long and several spectra of MIPAS are already available in the literature. Nevertheless, to satisfy the referee, we have included a new figure. It shows one spectrum in channel A near 40 km (this channel in mainly used below 50 km); and 3 spectra at tangent heights near 50, 60 and 90 km from channel AB, mainly used to retrieve O3 above 50 km.

-p.7 Line 10: What is the major improvement of the new version 5 of L1b spectra compared to the previous one (particularly compared to v-4.61/62)?

The upgrades in the Level 1b products for version 5 include both scientific and format updates. In particular: i) a truncation of the interferogram to 8.0 cm in order to avoid under-sampling the spectrum for the Optimized Resolution mission; ii) improved Level 1b engineering heights calculation ; iii) Calculation of the quadratic terms for spectral calibration that are provided in the output products; and iv) Additional fields in the Level 1b products, such as the auxiliary L0 data packets that provide information about house keeping data. More info is given at
https://earth.esa.int/documents/700255/707722/MIP_NL_1P_Disclaimers.pdf/17ae8d2b-f1ee-49a8-ade3-1bda7a7c1d7c

-p.9 Line 4: "...the calculation of the spectra the contribution (as a...": this sentence appears odd.

Right. It has been changed to: "The forward model also includes the contribution (as a potential overlap with O3 lines) of CO2 lines.".

-p.10 Line 11: Threshold of the averaging kernel 0.03, is this an empirical value?

This is based on the many tests we have carried out to characterize the retrieval performance. Retrieved values with an AK smaller that that essentially contains only a priori information.

-p.12 Error analysis for the systematic errors: I would suggest to add a short description about how the authors evaluated the systematic errors (numerically computed by comparing the retrieved profiles using the nominal inversion model and the modified inversion model?).

Correct. We have included a sentence in the middle of Sec. 4, when we start discussing the systematic errors.

-p.16 Line 30: Baron et al. (2011) seems a reference paper for the earlier version of SMILES O3 data.

Correct. That line has been removed.